# Thermodynamic phase diagram of the competition between superconductivity and charge order in cuprates

Giulia Venditti[1*], Ilaria Maccari[2], Jose Lorenzana[3†] and Sergio Caprara[3]

**1** SPIN-CNR Institute for Superconducting and other Innovative Materials and Devices, Area della Ricerca di Tor Vergata, Via del Fosso del Cavaliere 100, 00133 Rome, Italy
**2** Department of Physics, Stockholm University, Stockholm SE-10691, Sweden
**3** ISC-CNR and Department of Physics, Sapienza University of Rome, Piazzale Aldo Moro 2, 00185, Rome, Italy
*giulia.venditti@spin.cnr.it, †jose.lorenzana@cnr.it

July 25, 2023

## Abstract

We argue that there is a special doping point in the phase diagram of cuprates, such that the condensation of holes into a charge-ordered and into a superconducting phase are degenerate in energy but with an energy barrier in between. We present Monte Carlo simulations of this problem without and with quenched disorder in two-dimensions. While in the clean case, charge order and superconductivity are separated by a first-order line which is nearly independent of temperature, in the presence of quenched disorder, charge order is fragmented into domains separated by superconducting filaments reminiscent of the supersolid behaviour in $^4$He. Assuming weak interlayer couplings, the resulting three-dimensional phase diagram is in good agreement with the experiments.

# 1    Introduction

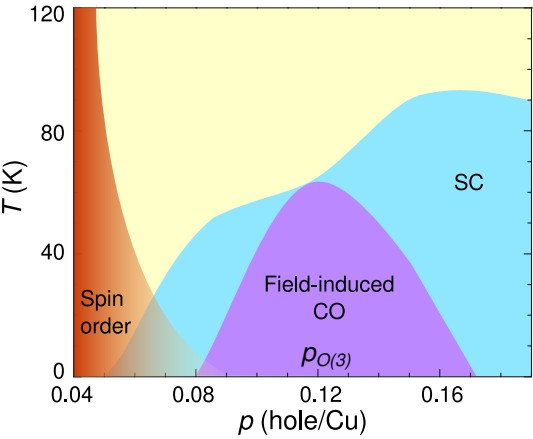

Figure 1: a) Phase diagram of $YBa_2Cu_3O_y$ in the temperature vs. hole doping plane (the figure is adapted from Ref. [1]). The magenta region is the CO induced by a magnetic field so the phase diagram can be seen as a two-dimensional projection of a three-dimensional phase diagram with the magnetic field axis running perpendicular to the plane of the figure. $P_{O(3)}$ indicates the "$O(3)$" doping at which CO and SC are nearly degenerate.

There is an overwhelming experimental evidence [2–9] that competition between charge order (CO) and superconductivity (SC) occurs in high-critical-temperature superconducting cuprates. It has been argued [8–12] that, under certain circumstances, the superconducting order parameter with $U(1)$ symmetry and a commensurate charge-density-wave (CDW) parameter with $Z_2$ (Ising) symmetry can be encoded in a single order parameter with approximate $O(3)$ symmetry. Evidence for this emergent symmetry stems from studies where the balance between SC and CO is controlled by a non-thermal parameter which couples non-linearly to one of the orders [4, 5, 7–9, 13, 14]. For example, a uniform magnetic field $H$ disfavours SC with respect to CO. At zero magnetic field, upon reducing the temperature, the correlation length of CO starts to grow at a temperature $T_{QC}$ larger than the superconducting critical temperature $T_c$, as if the system was approaching a charge-ordered state [15]. With further lowering the temperature, however, the growth of the CO correlation-length stops near $T_c$, where no CO but rather SC develops. Extrapolating the divergence of the CO correlation length to the superconducting region shows that the temperature $T_{CO}$ of the putative charge-ordered state coincides with the actual three-dimensional ordering temperature $T_{CO}^{3D}$ that is reached once SC is suppressed by a sufficiently strong magnetic field. One fact that points to an approximate $O(3)$ symmetry between the two orders is that the critical temperature for the field-induced CO (and therefore the putative charge-ordering temperature) obeys $T_{CO}^{3D} \approx T_c$ near a hole doping content, hereafter "the $O(3)$ point", $p_{O(3)} \approx 0.12$. As schematically shown in Fig. 1, this has been seen in $YBa_2Cu_3O_{6+x}$ with various probes as nuclear magnetic resonance [1],

sound velocity, and Hall effect measurements (see Ref. [16], and references therein). The degeneracy of the ordering temperature at the $p_{O(3)}$ point strongly suggests that the tendency towards CO and SC are the same instability manifesting in different channels.

It is useful to visualize this phase diagram with an extra control-parameter axis, such as the magnetic field, perpendicular to the $T$ vs. doping plane. Fig. 2 shows various cuprate experimental phase diagrams (a-c) and compare them with $^4$He (d). In panels (a) and (b), the control parameter is the magnetic field, which tunes the SC energy. We will refer to this situation as a SC-driven transition. In contrast, in panel (c) the control parameter is the isoelectronic doping, favouring stripes [17, 18], which will be referred to as a CO-driven transition. The superconducting and charge-ordered phases meet the disordered phase at a so-called "bicritical" point [19], analogous to the bicritical point in $^4$He (d). An alternative terminology is that of triple point, usually adopted when the transition lines are first-order, as for instance the point where the liquid, gas, and solid phases of a substance meet.

In all the phase diagrams of Fig. 2 (a-c), a superconducting foot develops underneath the charge-ordered state at low temperatures [wavy shading in (a-c)], which in Refs. [8, 9] was associated with the occurrence of filamentary SC (FSC). This is attributed to a *tertius gaudens* (rejoicing third) effect. The quenched disorder breaks the CO into domains hosting different variants of CO related by discrete translations. At the interface between two variants of CO, both get frustrated and SC is stabilized. The same principle is believed [22] to explain the appearance of supersolid phases in $^4$He [wavy shading in (d)].

A striking characteristic of the phase diagrams in Fig. 2 is the nearly vertical separation between CO/solid and SC/superfluid phases in all the phase diagrams, once FSC is disregarded. In cuprates, several probes near $p_{O(3)}$ show that, in a wide temperature range, the critical magnetic field $H_c^*$, at which a long-range charge-ordered phase stabilizes, is nearly independent of the temperature. This is seen in sound-velocity data [4], resonant inelastic X-ray scattering [14] and nuclear magnetic resonance [1, 7] experiments on $YBa_2Cu_3O_{6+x}$, as shown in Fig. 2(a). The transition to the superconducting phase has been determined by the anomaly in the density of states probed by specific heat measurements, which coincides with $T_c$ determined from other methods (see Ref. [7], and references therein). A similar phase diagram can be deduced from magnetotransport experiments [8, 9, 23, 24] in La-based cuprates. Here, the lines are not sharp anomalies, probably due to stronger disorder, but the general topology is the same [see Fig. 2(b)].

Applying a magnetic field is not the only way to tune the balance between CO and SC. An alternative path is the structural enhancement of CO introduced by Tranquada and collaborators [25]. In this case, isovalent doping induces a structural distortion which couples with CO. When plotted against the isovalent doping concentration [18, 26–28], the phase diagrams bear a striking resemblance with the magnetic-field-controlled phase diagrams for similar hole content [see Fig. 2(c)]. We notice, on passing, that the phase diagram of hole-doped La-based cuprates is characterized by a considerable (and unavoidable) coexistence of different structural phases, i.e., the low-temperature tetragonal (LTT) and the low-temperature orthorhombic (LTO) phases [29]. Here, the assigned filamentary phase is more prominent, which can be understood as the effect of higher disorder (cf. with Fig. 4(a) in Ref. [9]).

Notice that in panels (a) and (b) of Fig. 2, $T_c$ evolves rapidly with the control parameter, while the CO temperature is approximately constant. The situation reverses in panel (c). Clearly, this is due to the different way the control parameter couples to the two competing phases. In (a) and (b) the magnetic field destabilizes the superconducting phase having little influence on the CO. Instead, in (c) the structural distortion stabilizes the CO phase and has little effect on the superconducting $T_c$. In the case of $^4$He, pressure

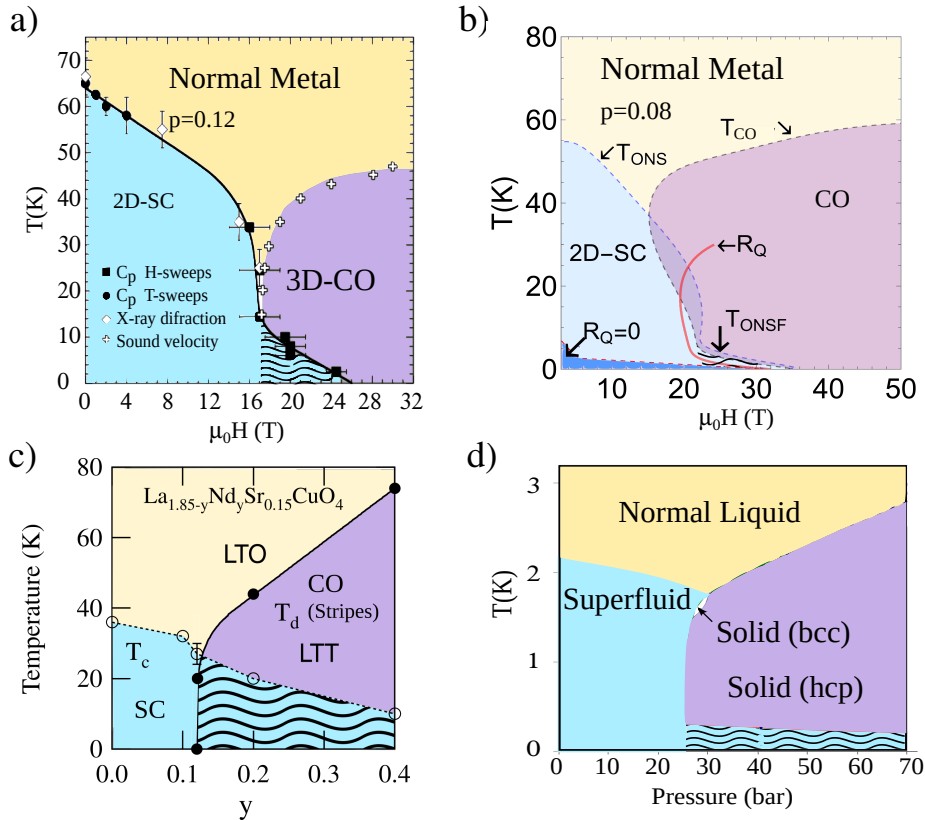

Figure 2: Phase diagrams showing competition between SC/superfluidity and CO/solid phases. a) $YBa_2Cu_3O_y$ with $y = 6.67$. The onset of superconducting correlations (labelled 2D-SC) was detected by an anomaly in the specific heat (adapted from Ref. [7]). Three-dimensional CO (3D-CO) was detected with X-ray diffraction [3] and sound velocity [4]. b) $La_{2-x}Sr_xCuO_4$ with $x = 0.08$. The onset temperatures for SC ($T_{ONS}$), FSC ($T_{ONSF}$), and CO ($T_{CO}$) were extracted from magnetoresistance data (adapted from Ref. [9]). The darker blue region corresponds to zero sheet resistance $R_\square = 0$. The red line $R_Q$ shows the locus of the quantum of resistance. c) $La_{1.85-y}Nd_ySr_{0.15}CuO_4$. Here, isoelectronic doping ($y$) favours the LTT, which stabilizes CO in the form of stripes [17]. Open circles show $T_c$ while solid circles show the structural transition $T_d$ which is known to be close to the CO temperature (adapted from Ref. [18]). d) Phase diagram of $^4$He (adapted from Ref. [20]). In a)-c) the wavy-shaded regions denote the coexistence of CO and SC which in Ref. [8, 9] was attributed to filamentary superconductivity. In d) the wavy shading is the $^4$He-analogous supersolid phase observed in Ref. [21].

stabilizes the solid phase, which indeed has a larger critical temperature slope.

Summarizing, the cuprate phase diagrams are in many respects similar to the $^4$He phase diagram shown in Fig. 2(d), namely: i) similar critical temperatures for SC/superfluid and CO/solid, ii) vertical transition lines and iii) FSC/supersolid phase induced by disorder and grain boundary effects.

For $^4$He, the control parameter is the pressure $P$. At low temperatures, and at a critical pressure nearly independent of the temperature, the superfluid phase transforms into the solid phase. The analogy between a continuum system like $^4$He and lattice systems as cuprates is not new. Indeed, at least from a theoretical point of view, there is a long tradition [30, 31] of modelling $^4$He on discrete lattices, very similar in spirit to the model we shall be using for cuprates in this work. Also, analogies between the phase diagram of cuprates and $^4$He had been emphasized before [32].

The vertical transition line $T_{SF\leftrightarrow S}(P)$ in (c) between the superfluid (subscript SF) and the solid phase (subscript S), more often drawn as horizontal with exchanged axes, was one of the first arguments put forward by London to advocate for some form of condensation in the early days of research on superfluids. Indeed, according to the Clausius-Clapeyron equation, relating the changes in entropy $\Delta S$ and in volume $\Delta V$ at a first-order phase transition, the divergent derivative $dT_{SF\leftrightarrow S}/dP = \Delta V/\Delta S$ implies that $\Delta S = 0$, thus identifying the superfluid as a practically zero-entropy state, like a solid phase [33]. Analogously, the divergent $dT_{SC\leftrightarrow CO}/dH = \Delta M/\Delta S$ (with $\Delta M$, the change in magnetization) in cuprates implies that at the critical field, the same quasiparticles flip from a momentum-condensed state to a real-space condensed state, that represents two equally ordered (low-entropy) states.

A minimal model to investigate the instability that can occur in the particle-hole or particle-particle channel, i.e., pre-formed electron pairs that are paired in real space (precursors of CO) or in momentum space (Cooper pairs, precursors of SC) is the two-dimensional attractive Hubbard model [34]. This model enjoys the property that, exactly at half-filling (one electron per unit cell), CO and SC are degenerate. Moving the electron density away from half-filling usually tilts the balance in favour of SC, unless other interactions (e.g., a nearest-neighbor repulsion) are present.

The attractive Hubbard model can be mapped onto the repulsive Hubbard model [34], which has a spin-density-wave ground state. In this representation, spin-density-wave order along $z$ describes a charge-ordered state, while order in the $xy$ plane describes SC. Order along any other direction maps into uniform "supersolid" order, where SC and CO coexist. Since the free energy of the repulsive model is invariant with respect to $O(3)$ rotations of the order parameter, one concludes that the charge-ordered, superconducting and supersolid phases are degenerate. In two spatial dimensions, this suppresses the ordering temperature to $T = 0$, due to the Mermin-Wagner theorem [35].

More insight into the competition between CO and SC can be gained considering the limit of a large Hubbard coupling, where the model can be mapped onto a Heisenberg model [34] of interacting pseudospins (analogous to Anderson's pseudospins). Here, the pseudospin projection along $z$, up or down, encodes a double occupied or an empty site on the lattice, respectively, while the in-plane component encodes superconducting correlations. While the Hubbard model is genuinely quantum, it is well known [36] that, provided the ground state is ordered above a microscopic scale (the Josephson correlation length) $\xi_J \approx \hbar c/J_s$, the system can be described by a classical theory. Here, $J_s$ and $c$ are the stiffness and the zero-temperature spin-wave velocity of the effective Heisenberg model, respectively. Using estimates appropriate for the latter [36], one obtains $\xi_J = 2\pi a/C_s \approx 10.9\,a$ with $C_s$ a constant, and $a$ the lattice spacing. Therefore, we can use a classical effective lattice spin model to study the competition between CO and SC. Each

pseudospin in the lattice model represents a cluster of elementary unit cells with a linear dimension of order $\xi_J$ behaving as a classical variable. The superconducting transition in two dimensions belongs to the Berezinskii–Kosterlitz–Thouless (BKT) universality class, and is characterized by the appearance of a finite stiffness and the binding of vortices and antivortices.

While the above scenario is very appealing to formulating a statistical mechanical description of the competition between CO and SC, a full $O(3)$ symmetry is clearly a drawback of the model. Indeed, for the repulsive Hubbard model, this is a consequence of rotational invariance, but there is no such fundamental symmetry in a generic attractive model. One expects that CO and SC can be tuned to an approximate $O(3)$ symmetry point by a non-ordering field (e.g. $p$ for cuprates), but there is no reason why the barrier between these states should vanish at the $O(3)$ point. In other words, the $O(3)$ symmetry is only approximate, in the sense that the charge-ordered and superconducting phases are still degenerate but are separated by barriers.

Based on these considerations, we study an effective classical spin model on a square lattice, with nearest neighbour exchange interaction and three relevant parameters: an exchange anisotropy, to tilt the balance between easy-axis (charge) and easy-plane (superconducting) order; a potential barrier, to remove the unphysical high degeneracy of the $O(3)$ symmetric point; a random field to mimic disorder.

In the clean system (without disorder), we find that the presence of the barrier allows for a finite-temperature phase transition, otherwise forbidden at the $O(3)$ point. Once disorder is taken into account, CO is fragmented into different domains, resulting in a polycrystalline charge-ordered phase, and FSC sets in as a parasitic phase at the domain boundaries [8,9].

Our analysis is carried out by means of Monte Carlo (MC) calculations, which allow us to study not only the ground state, as in Ref. [9], but the thermodynamic phase diagram itself and the behaviour in temperature of the various physical quantities.

The scheme of this work is the following. In Sec. 2, we discuss the model and methods of investigation. In Sec. 3, we discuss the properties of the model in the absence of disorder, highlighting the role of the potential barrier and using the result of the model in the absence of a barrier [37–39] as a benchmark. The phase diagrams for small and large values of the potential barrier are discussed in detail in Secs. 3.3 and 3.4, respectively. In Sec. 4, we include the effect of disorder and show that this is crucial to promote FSC. Our concluding remarks are found in Sec. 6.

## 2  Model and Methods

Above the Josephson scale, we can model our system with a classical order parameter. Therefore, as in Ref. [8–10] we consider a coarse-grained model of classical pseudospin vectors $\mathbf{S}_{\boldsymbol{R}}$ on the sites $\boldsymbol{R}$ on a square lattice, each representing a region of area $\xi_J^2$ of the quantum system. The new (coarse-grained) lattice spacing is set as $a' = 1$ and the linear size is $L$ (i.e., the lattice hosts $N = L^2$ sites), with periodic boundary conditions. The states with positive or negative pseudomagnetization along the $z$ axis represents two variants of the charge-ordered state, related in the original quantum microscopic model by a translation symmetry, while the in-plane pseudomagnetization describes the superconducting state [8–10]. In order to lighten the notation, we will henceforth refer to the pseudomagnetization simply as magnetization, not to be confused with the physical magnetization mentioned in connection with the Clausius-Clapeyron argument above. In the following, we set $|\mathbf{S}_{\boldsymbol{R}}| = 1$, and fix the reference frame so that the three Cartesian

components of the vector $\mathbf{S_R}$ are $S_R^x = \sin\varphi_R\cos\theta_R$, $S_R^y = \sin\varphi_R\sin\theta_R$, and $S_R^z = \cos\varphi_R$, in terms of the polar and azimuthal angles $\theta_R$ and $\varphi_R$. $\theta_R$ can be identified with the phase of the superconducting order parameter.

The competition between CO and SC is captured by the classical anisotropic Heisenberg model (XXZ model) with an effective barrier potential term and a random field mimicking disorder,

$$H = -J \sum_{\langle \mathbf{R},\mathbf{R'}\rangle} \left(S_R^x S_{R'}^x + S_R^y S_{R'}^y + \alpha\, S_R^z S_{R'}^z\right) + 4B\sum_{\mathbf{R}} (S_R^z)^2 \left[1 - (S_R^z)^2\right] + \frac{W}{2}\sum_{\mathbf{R}} h_R S_R^z,$$

(1)

where the symbol $\langle \mathbf{R},\mathbf{R'}\rangle$ specifies that the sum runs over nearest-neighbouring sites. We fix the interaction strength $J = 1$ unless otherwise specified. Furthermore, we use the anisotropy parameter $\alpha \geq 0$ to tune the ground state from being superconducting to charge-ordered. This corresponds to keep constant the ground-state energy of the SC and tune the one of the CO i.e. a CO-driven transition. As we shall see, a simple rescaling of the energy units allows describing a SC-driven transition.

The second term in Eq 1 is the barrier potential, whose height is adjusted by the parameter $B$. Its role, as anticipated in Sec. 1, is to eliminate the unphysical degeneracy of the charge-ordered and superconducting state with all possible intermediate supersolid phases for $\alpha = 1$. In the following, we will still call the $\alpha = 1$ case "the $O(3)$" or "isotropic" point, keeping in mind that such terminology refers only to the first term of the Hamiltonian in Eq. 1.

The last term in Eq. (1) is a random field that mimics impurities coupled to the charge density in a real system. We take $h_R$ as independent random variables with a flat probability distribution between $-1$ and $+1$. The strength of disorder is controlled by the parameter $W$. As we shall show, this term is crucial to promote the polycrystalline behaviour of CO for $\alpha \gg 1$, as well as the occurrence of FSC in a certain range of anisotropy $\alpha \gtrsim 1$, in the form of topologically protected domain walls between regions hosting two different realizations of CO.

In the case where $B = W = 0$, Eq. (1) is the bare XXZ model which has been widely studied in the literature [37–39], and whose phase diagram will be used as a benchmark case when discussing the effect of the energy barrier. In the bare model, the anisotropy $\alpha$ allows switching from the BKT universality class, for $\alpha < 1$, where the ground state is superconducting, to the Ising universality class, for $\alpha > 1$, where it is a charge-ordered ground state. Finally, in the isotropic limit $\alpha \to 1^{+,-}$, the critical temperature goes to zero logarithmically [40], and at $\alpha = 1$ no finite-temperature phase transition is possible, according to the Mermin-Wagner theorem [35].

The presence of a finite energy barrier separating the three equivalent ground states $\varphi_R = 0, \pi/2, \pi$ (i.e., $S_R^z = -1, 0, 1$) makes the model no longer invariant with respect to $O(3)$ rotations of the order parameter, and the Mermin-Wagner theorem does not apply. Indeed, we find that, for $B > 0$, the ordering temperature remains finite for all $\alpha$. Furthermore, since the effect of the barrier persists at finite temperatures, metastability regions appear in the resulting $T$ vs $\alpha$ phase diagram.

In order to study the physical quantities related to the effective Hamiltonian, Eq. (1), as functions of the temperature, we performed large-scale Monte Carlo (MC) simulations, with systems of linear size $L$ ranging from $L = 16$ up to $L = 256$. We used the Metropolis and simulated annealing algorithms to optimize the thermalization process: at the highest temperature reached in our calculations the system evolves from an initial configuration of random pseudospins until it reaches its equilibrium state, then the temperature is slightly decreased and a new thermalization starts from the final configuration of the previous step. This process is iterated until the lowest temperature of interest is reached. At each

Metropolis step, the whole lattice is updated according to the Metropolis prescription [41], either sequentially updating all the $L \times L$ pseudospins or by $L \times L$ random choices of the pseudospin to be updated. Thermal averages of any observable $\mathcal{O}$ are obtained as the average over $N_{\text{MC}}$ (at least $10^3$) measures,

$$\langle \mathcal{O} \rangle = \frac{1}{N_{\text{MC}}} \sum_{j=1}^{N_{\text{MC}}} \mathcal{O}_j,$$

taken $\tau_{\text{MC}}$ Metropolis steps apart from one another, $\tau_{\text{MC}}$ being of the order or larger than the autocorrelation time (for the clean system, typically we take $\tau_{\text{MC}} = 30 - 100$ Metropolis steps). To account for the thermalization time, we finally discard the initial $N_{\text{out}}$ (at least $10^5$) Metropolis steps. In the presence of the random field, we also average over $N_{\text{dis}}$ realizations of the disorder (henceforth, this average is marked by an overline).

## 2.1 Physical observables

To analyse the tendency to order, we compute the mean-square magnetization,

$$\widetilde{\chi}^{\nu} \equiv N \langle m_{\nu}^2 \rangle = \frac{1}{N} \left\langle \left( \sum_{\boldsymbol{R}} S_{\boldsymbol{R}}^{\nu} \right)^2 \right\rangle, \quad \nu = x, y, z, \tag{2}$$

where $m_{\nu} = \frac{1}{N} \sum_{\boldsymbol{R}} S_{\boldsymbol{R}}^{\nu}$ is the magnetization per unit surface area calculated at each MC step. Note that the mean-square magnetization is directly related to the charge ($\nu = z$) and superconducting $\nu = x, y$ susceptibilities [39],

$$\chi^{\nu} = \frac{1}{T} \left[ \left\langle \left( \sum_{\boldsymbol{R}} S_{\boldsymbol{R}}^{\nu} \right)^2 \right\rangle - \left\langle \sum_{\boldsymbol{R}} S_{\boldsymbol{R}}^{\nu} \right\rangle^2 \right].$$

In the absence of long-range order $\langle m_{\nu} \rangle = 0$, therefore, for a BKT system in the thermodynamic limit, $\chi^{\nu} = N \widetilde{\chi}^{\nu}/T$. Of course, in numerical calculations, the system is always finite and never reaches the real thermodynamic limit, preventing the vanishing of $\langle m_{x,y} \rangle$. In the following, we will use the quantity $\widetilde{\chi}^{xy} \equiv \frac{1}{2}(\widetilde{\chi}^x + \widetilde{\chi}^y)$, to monitor the superconducting correlations. While $\widetilde{\chi}^{xy}$ can be seen as a proxy of the superconducting susceptibility, in the charge-ordered phase, instead, the order parameter is nonzero, so that $\chi^z$ and $\widetilde{\chi}^z$ are not simply proportional. In this case, to monitor the response of the CO correlations, we use both the susceptibility $\chi^z$, which is the true response of the system to an external field, and $\widetilde{\chi}^z$.

In order to assess a global BKT superconducting transition, we compute the in-plane superfluid stiffness $J_{\text{s}}$, associated with the superconducting phase rigidity and defined as the second derivative of the free energy with respect to a twist of the SC phase angle $\delta\theta$, e.g., along the $x$ direction:

$$J_{\text{s}}(L, T) = -T \frac{\partial^2 \ln Z(\delta\theta)}{\partial \delta\theta^2}\bigg|_{\delta\theta=0} =$$

$$= \frac{1}{L^2} \left\langle \sum_{\boldsymbol{R}} \sin \varphi_{\boldsymbol{R}} \sin \varphi_{\boldsymbol{R}+\hat{\boldsymbol{x}}} \cos(\theta_{\boldsymbol{R}} - \theta_{\boldsymbol{R}+\hat{\boldsymbol{x}}}) \right\rangle$$

$$- \frac{1}{L^2 T} \left\langle \left( \sum_{\boldsymbol{R}} \sin \varphi_{\boldsymbol{R}} \sin \varphi_{\boldsymbol{R}+\hat{\boldsymbol{x}}} \sin(\theta_{\boldsymbol{R}} - \theta_{\boldsymbol{R}+\hat{\boldsymbol{x}}}) \right)^2 \right\rangle$$

where $Z$ is the partition function and $\hat{\boldsymbol{x}}$ is the unit vector in the $x$ direction.

To perform the extrapolation to the thermodynamic limit of the BKT critical point we use the BKT scaling of the superfluid stiffness [42],

$$\frac{J_{\text{s}}(L, T_{\text{BKT}})}{1 + [2\ln(L/L_0)]^{-1}} = \frac{2}{\pi} T_{\text{BKT}}, \tag{3}$$

where $T_{\text{BKT}}$ is the BKT critical temperature and we take $L_0$ as a fitting parameter.

Whenever needed, we complement our analysis of the superconducting state, with a closer inspection into the occurrence of vortices in the pattern of the local superconducting order parameter (the in-plane magnetization). A vortex (anti-vortex) is identified whenever a variation of $2\pi$ $(-2\pi)$ of the superconducting phase $\theta_{\boldsymbol{R}}$ is found in a closed path around a single plaquette with side equal to the lattice spacing. Defining the superconducting phase difference at site $\boldsymbol{R}$ in the direction of the $\hat{\boldsymbol{\nu}}$ unit vector ($\hat{\boldsymbol{\nu}} = \hat{\boldsymbol{x}}, \hat{\boldsymbol{y}}$) as

$$\Theta_{\hat{\boldsymbol{\nu}}}(\boldsymbol{R}) = [\theta_{\boldsymbol{R}} - \theta_{\boldsymbol{R}+\hat{\boldsymbol{\nu}}}]_{-\pi}^{+\pi},$$

the notation $[\cdot]_{-\pi}^{+\pi}$ meaning that we take the value modulus $2\pi$ so that $\Theta_{\hat{\boldsymbol{\nu}}}(\boldsymbol{R}) \in (-\pi, \pi]$, the circulation of the superconducting phase around a plaquette whose center is located at $\boldsymbol{R} + \frac{1}{2}(\hat{\boldsymbol{x}} + \hat{\boldsymbol{y}})$ is

$$\Theta_{\hat{\boldsymbol{x}}}(\boldsymbol{R}) + \Theta_{\hat{\boldsymbol{y}}}(\boldsymbol{R} + \hat{\boldsymbol{x}}) - \Theta_{\hat{\boldsymbol{x}}}(\boldsymbol{R} + \hat{\boldsymbol{y}}) - \Theta_{\hat{\boldsymbol{y}}}(\boldsymbol{R}) = 2\pi n_{\boldsymbol{R}},$$

where $n_{\boldsymbol{R}} = \pm 1$ is the integer vorticity in the phase angle $\theta_{\boldsymbol{R}}$ going around the plaquette [43]. Summing over all positive (negative) vorticities per unit length we obtain the density of vortices (antivortices) $\rho_{\text{V}} > 0$ ($\rho_{\text{AV}} < 0$), defining the total vorticity as

$$\rho_{\text{V, tot}} = \rho_{\text{V}} - \rho_{\text{AV}}. \tag{4}$$

Concerning the charge-ordered state, we define $T_{\text{CO}}$ using the crossing point (as a function of temperature) of the kurtosis of the pseudospin distribution function, i.e., the Binder cumulant

$$U_N = 1 - \frac{\langle m_z^4 \rangle}{3\langle m_z^2 \rangle^2}. \tag{5}$$

Its value at the critical temperature is indeed less sensitive to finite-size effects, as compared to the CO order parameter $\langle m_z \rangle$, and unbiased by fitting functions and *a priori* scaling hypotheses. In the thermodynamic limit $N \to \infty$, one expects $U_N \to 0$ in the high-temperature limit, while $U_N \to 2/3$ in the ordered phase for $T \to 0$ [44–46].

# 3 Clean system

## 3.1 Metastability and spinodal lines

Let us start discussing the MC numerical results starting from the clean case in the presence of a finite energy barrier, i.e. $B > 0$. As can be expected, one prominent effect of the barrier is to introduce metastability in the system. Already at zero temperature, there exists a range of values of $\alpha_{\text{CO}}^*(B) < \alpha < \alpha_{\text{SC}}^*(B)$ where both phases are local minima of the energy [47]. The spinodal points $\alpha_{\text{CO,SC}}^*(B)$ mark the limit of stability of the less stable phase (i.e., CO, decreasing $\alpha$).

At zero temperature, the spinodal points $\alpha_{\text{CO,SC}}^*(B)$ can be calculated analytically as a function of the barrier height $B$. Indeed, increasing $\alpha$ at $T = 0$, we can assume that

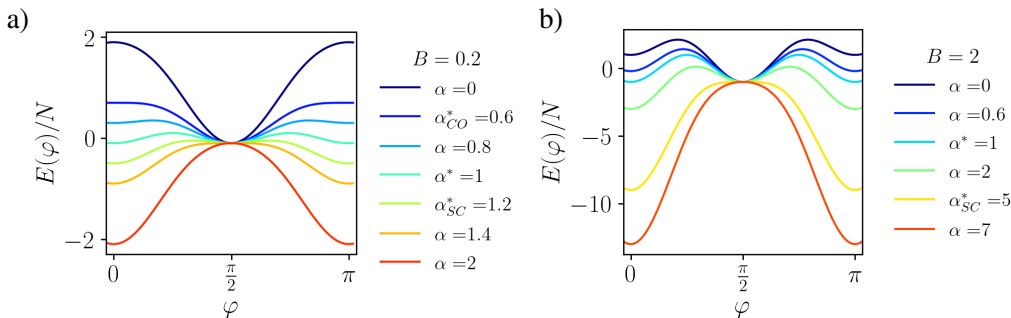

Figure 3: Energy landscapes [Eq. (6)] at $T = 0$, for various $\alpha$ and: a) $B = 0.2$; b) $B = 2$. Notice that in this case even for $\alpha = 0$ the charge-ordered phase remains as a metastable minimum, and there are two possible realizations (two equivalent minima).

all pseudospins are parallel, i.e., $\varphi_{\boldsymbol{R}} = \varphi$ and $\theta_{\boldsymbol{R}} - \theta_{\boldsymbol{R}'} = 0$. Up to constant terms in the angle $\varphi$, the total energy per site from Eq. (1) takes the simplified form

$$\frac{1}{N} E(\varphi) = (1 - \alpha) \cos(2\varphi) - \frac{1}{2} B \cos(4\varphi). \tag{6}$$

The resulting energy landscapes for $B = 0.2$ and $B = 2$, with varying $\alpha$, are given in Figs. 3(a) and (b), respectively. The free energy has a single superconducting global minimum at $\varphi = \pi/2$, up to some $\alpha = \alpha^*_{\text{CO}}(B)$, after which two new *local* (metastable) minima at $\varphi = 0, \pi$, corresponding to CO, appear. Crossing the isotropic point $\alpha^*$ the situation gets reversed: the new global minima are found at $\varphi = 0, \pi$ and the $\varphi = \pi/2$ configuration becomes a local (metastable) minimum, disappearing at some $\alpha^*_{\text{SC}}(B)$, after which only the two equivalent charge-ordered states survive.

By substituting $\varphi = \pi/2$ and $\varphi = 0, \pi$ in the second derivative of Eq. (6), we obtain the two spinodal points at zero temperature,

$$\alpha^*_{\text{SC,CO}}(B) = 1 \pm 2B. \tag{7}$$

For $B = 0$ the model becomes fully symmetric and the zero-temperature spinodal points merge into the zero-temperature transition point.

The study of metastable states at finite temperatures is particularly challenging, both in real experiments and in numerical simulations. If the system is prepared in a metastable state, there is a finite probability that a bubble of the more stable phase nucleates in a finite time and then grows. Therefore, sooner or later the system will transit to the more stable phase. As a consequence, spinodal lines, marking the limit of stability of the metastable phase, require time-domain considerations to be defined. A metastable phase is well-defined only if it persists at least for its equilibration time, otherwise, it can not be considered a thermodynamic phase. The line where the equilibration and the nucleation times become equal defines the dynamical spinodal line [48]. While these considerations allow a rigorous definition of spinodals, here we circumvent the problem of estimating these times and take a more pragmatic approach which is enough for our purpose, as follows. For a small enough system, during a simulation, the system may get trapped in one metastable state for a long time (compared to the equilibration time of the state) and sporadically change to a different state, where it gets trapped again. Evolving the system long enough, we can construct a reliable effective probability density function $P_{\text{eff}}(m_z)$ for the order parameter associated with CO, that is encoded in $m_z = \frac{1}{N} \sum_{\boldsymbol{R}} S^z_{\boldsymbol{R}}$. By definition, the free energy of the system is $F(m_z) = -T \ln[P_{\text{eff}}(m_z) / \sum_{m_z} P_{\text{eff}}(m_z)]$. Notice that by doing

the histogram in $m_z$ one automatically takes into account the correct measure for the probability distribution. It is easy to check that, for $B = 0$, the probability distribution is flat at $\alpha = 1$, $P(m) = \frac{1}{2}$, consistent with a flat free energy.

The numerical identification of the spinodal lines and the equilibrium coexistence line of SC and CO can be obtained by studying the form of $F(m_z)$ as a function of the parameters. For a fixed value of $\alpha$, the first-order transition temperature $T_{SC \leftrightarrow CO}$, is defined as the temperature at which the absolute minimum of the free energy changes from $m_z = \pm 1$ to $m_z = 0$. A spinodal temperature $T_{\rm sp}$ is defined by the local minimum of a metastable phase becoming an inflection point. Thus, once we extracted all the effective free energies $F(m_z)$ at each temperature, we can infer the full phase diagram.

Since the flip of the whole phase is a very rare event, we need to take a very small system in order to have enough flips to consider the system at equilibrium within a reasonable simulation time. As a proof of principle and in order to have an approximate map of the spinodal lines, we take $L = 4$ and construct $P_{\rm eff}(m_z)$ using histograms of $m_z$ measured at each MC step. The system is evolved for $N_{\rm MC} = 5 \times 10^5$ after $N_{\rm out} = 5 \times 10^5$, with $\tau_{\rm MC} = 50$. Of course, it must be borne in mind that the condition $m_z = 0$ cannot distinguish a superconducting state from a charge-disordered state. To construct the phase diagram one has to complement the previous study with the computation of the superfluid stiffness.

The search for metastable states and first-order lines within the above protocol becomes harder and harder with increasing $B$. Thus, the investigation of the metastable states must be adapted to the cases of a small ($B \ll 1$) and a large ($B \gg 1$) barrier. Henceforth, we shall refer to the paradigmatic cases $B = 0.2$ and $B = 2$ to discuss the two different regimes.

In the case of large $B$, we follow a different protocol to numerically estimate the spinodal points. For $\alpha < \alpha_{\rm B}$ ($\alpha > \alpha_{\rm B}$) the superfluid stiffness (Binder cumulant) are computed starting from a charge-ordered (superconducting) configuration and heating up the system, using the simulated annealing algorithm. We define a spinodal point using the temperature at which $J_{\rm s}$ or $\langle m_z^2 \rangle$ jump to their finite value, checking that this temperature is not strongly dependent on the system size $L$. The absence of a significant size dependence can also be viewed as a confirmation of the spinodal points extracted with the free energy protocol.

In Sec. 3.2 we characterize bicriticality within our model, whereas in Secs. 3.3 and 3.4 we discuss in detail two paradigmatic examples of phase diagrams, for a small and large value of the barrier height, respectively.

## 3.2 Bicriticality

According to the Mermin-Wagner theorem [35], the isotropic Heisenberg model, i.e., Hamiltonian (1) with $W = B = 0$ and $\alpha = 1$ has no long-range order. We discussed already in Sec. 2 the unphysical nature of the full $O(3)$ symmetry that characterizes the isotropic point without a barrier. We will show in Secs. 3.3 and 3.4 that the bicritical point $(\alpha_{\rm B}, T_{\rm B})$, at which SC, CO and the disordered phase meet, gets shifted to $\alpha_{\rm B} > 1$.

To gain a first insight into this phenomenon, we compare in Fig. 4 the superconducting $\widetilde{\chi}^{xy}$ and charge-ordered $\widetilde{\chi}^z$ functions, defined as in Eq. (2) for different values of $B > 0$. When the barrier is present ($B > 0$), we observe a sizable $\widetilde{\chi}^{xy}$. The temperature at which the superconducting response significantly rises is an increasing function of $B$ (see Fig. 4a). At lower temperatures, the mean-square magnetization tends to a finite value, indicating the stabilization of superconducting correlations. Indeed, contextually, $\widetilde{\chi}^z$ presents a peak and is driven to zero at low temperatures. This behaviour is characteristic of the bare XXZ model with $\alpha < 1$, i.e., in the superfluid region of the phase diagram. In the presence

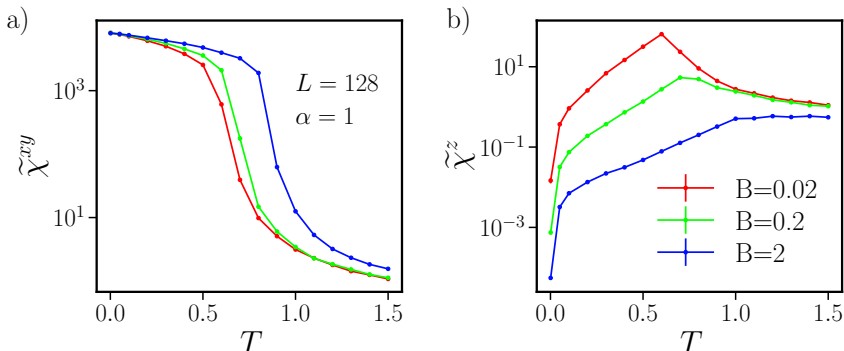

Figure 4: a) Superconducting $\widetilde{\chi}^{xy}$ and b) charge-ordered $\widetilde{\chi}^{z}$ mean-square magnetizations vs. temperature $T$ for the isotropic case $\alpha = 1$ at various heights of the potential barrier $B$. While for the bare Heisenberg model ($B = 0$) no transition is possible, the presence of a barrier allows for a BKT transition at $\alpha = 1$. We used $L = 128$, $N_{MC} = 2 \times 10^3$, $N_{out} = 5 \times 10^4$ and $\tau_{MC} = 100$. The error bars are calculated using the bootstrap resampling method with 100 datasets and blocks of size 100 [49].

of the barrier, we find that the same results also persist for a small range of $\alpha > 1$. Thus, the effect of the barrier is to shift the bicritical point $(\alpha_B, T_B)$ to $\alpha_B > 1$. However, as discussed in the previous section, at $T = 0$ the superconducting and the charge-ordered phases are degenerate at $\alpha = 1$. This implies that the first order line $T_{CO \leftrightarrow SC}$, which by definition starts at $(\alpha = \alpha^* \equiv 1,\ T = 0)$ and ends at the bicrtical point, must have a positive slope. This indicates that for a small range $\alpha \gtrsim 1$ and lowering the temperature, one has the sequence of phases: disorder $\to$ SC $\to$ CO. Thereby, two spinodal lines starting from $(\alpha_B, T_B)$ and terminating at points $\alpha^*_{CO, SC}(B)$ and $T = 0$ should appear, as will be illustrated in the next sections.

## 3.3 Phase diagram for $B = 0.2$

In this section, we discuss the case $B = 0.2$. In Fig. 5 we plot the superconducting $\widetilde{\chi}^{xy}$ (panel a) and charge-ordered $\widetilde{\chi}^{z}$ (panel b) mean-square magnetization, as well as the susceptibility $\chi^{z}$ (panel c), for different values of the anisotropy parameter, in the range $0.1 < \alpha < 2$.

The superconducting mean-square magnetization $\widetilde{\chi}^{xy}$ grows monotonously by lowering the temperature for values of the anisotropy parameter as large as $\alpha = 1.04$, i.e. above the isotropic Heisenberg limit. For the same range of anisotropy, $\widetilde{\chi}^{z}$ shows a maximum and then drops nearly to zero at lower temperatures. Clearly, thus, in this region the superconducting phase prevails at low temperature, so that, at some temperature $T_{BKT}$, the system transitions from a high-temperature disordered state to a superconducting state.

For $\alpha > 1.04$, the situation gets reversed with the charge correlations growing monotonically and the superconducting ones getting suppressed. This behaviour is coherent with the results found in the anisotropic Heisenberg model without a barrier ($B = 0$), where $\widetilde{\chi}^{z}$ decreases with $T$ for $\alpha \ll 1$ while for $\alpha \lesssim 1$ it displays a peak at $T \simeq T_{BKT}$ [39], as a precursor of the Ising transition that is found for $\alpha > 1$.

Observing the yellow curve corresponding to $\alpha = 1.04$, in Fig.5a, Fig.5b, and Fig.5c, one can understand the importance of considering all three quantities $\widetilde{\chi}^{xy}$, $\widetilde{\chi}^{z}$ and $\chi^{z}$. As

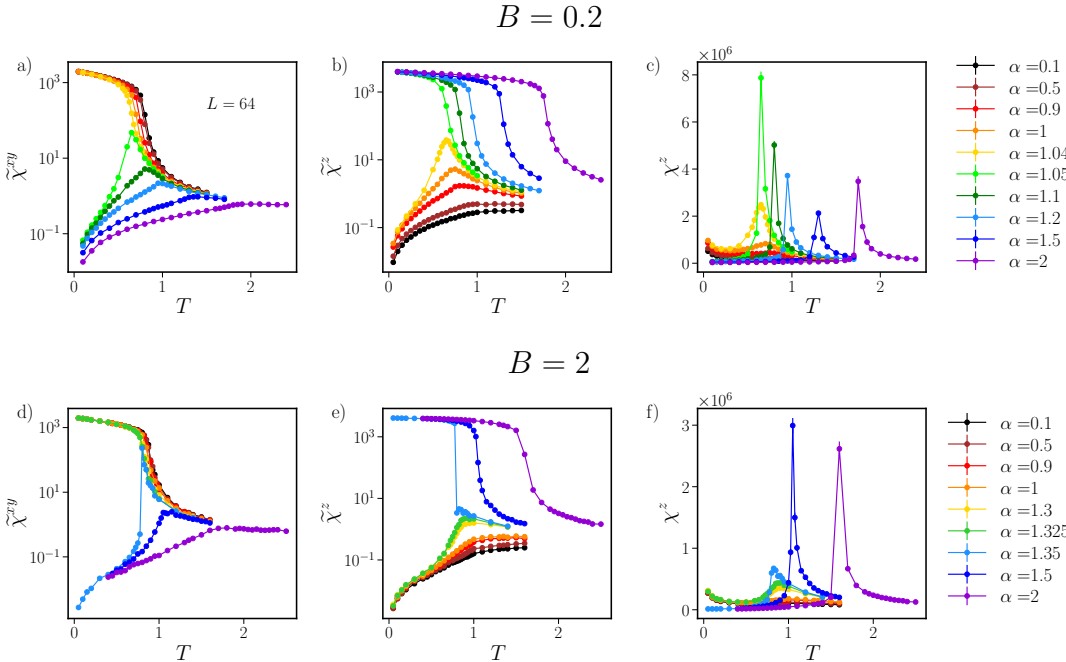

Figure 5: a) $\widetilde{\chi}^{xy}$, b) $\widetilde{\chi}^{z}$, and c) $\chi^{z}$, at different values of the anisotropy parameter $\alpha$, for a barrier parameter $B = 0.2$. We used $L = 64$, $N_{MC} = 10^4$ and $\tau_{\mathrm{MC}} = 30$ (50) discarding the first $6\times10^5$ ($2.5\times10^6$) MC configurations when $\alpha \leq 1$ ($\alpha > 1$). d) $\widetilde{\chi}^{xy}$, e) $\widetilde{\chi}^{z}$, and f) $\chi^{z}$ at different values of the anisotropy parameter $\alpha$, for a barrier parameter $B = 2$. Parameters are the same except that $\tau_{\mathrm{MC}} = 40$ and $N_{\mathrm{out}} = 8 \times 10^5$. The error bars are calculated using the bootstrap resampling method with 100 dataset and blocks of size 100.

a matter of fact, the CO susceptibility $\chi^z$ in Fig.5c presents a peak at $\alpha = 1.04$, although smeared with respect to the peaks for $\alpha \geq 1.05$; concurrently, $\widetilde{\chi}^{xy}$, Fig.5a, shows that a superconducting state is present at $\alpha = 1.04$. The doubt about whether the system has a superconducting or charge-ordered ground state is solved by looking at $\widetilde{\chi}^z$, Fig.5b, in which the $\alpha = 1.04$ curve grows with $T$ following the typical behaviours of the charge-ordered states $\alpha \geq 1.05$, but then decreases below a temperature $\widetilde{T} = 0.65$, at which $\widetilde{\chi}^z \sim 37$. The BKT scaling of $J_s$ for $\alpha = 1.04$ allowed us to extract $T_{\mathrm{BKT}} = 0.575 \lesssim \widetilde{T}$.

As soon as $\alpha \geq 1.05$, the main response of the system is in the out-of-plane direction (corresponding to CO), as it is clear looking at $\widetilde{\chi}^z$, Fig.5b, and at the susceptibility $\chi^z$, Fig.5c.

The resulting phase diagram is reported in Fig.6a. For comparison, we also report the phase diagram for $B = 0$ (light-blue line). Cyan circles and purple triangles refer, respectively, to the critical temperatures $T_{\mathrm{BKT}}$ and $T_{\mathrm{CO}}$ calculated using the scaling laws of $J_s$ [Eq. (3)] and $U_N$ [Eq. (5)]. The points in green along the line $T = 0$ are the analytical results: the two squares at $\alpha_{\mathrm{CO}}^* = 0.6$ and $\alpha_{\mathrm{SC}}^* = 1.4$ are the spinodal points, calculated as described in Sec. 3 [see Eq. (7)], and the green square at $\alpha^* = 1$ is the value at which the free energy has three equivalent minima at $m_z = 0, \pm1$ (first-order phase transition at $T = 0$).

As anticipated, the presence of the barrier with $B = 0.2$ shifts the superconducting transition line to higher temperatures, up to a value $\alpha_B = 1.04$ of the anisotropy control parameter. For slightly larger values, we recover the CO (Ising) transition line, which is shifted downwards with respect to the case $B = 0$. The bicritical point is shifted to

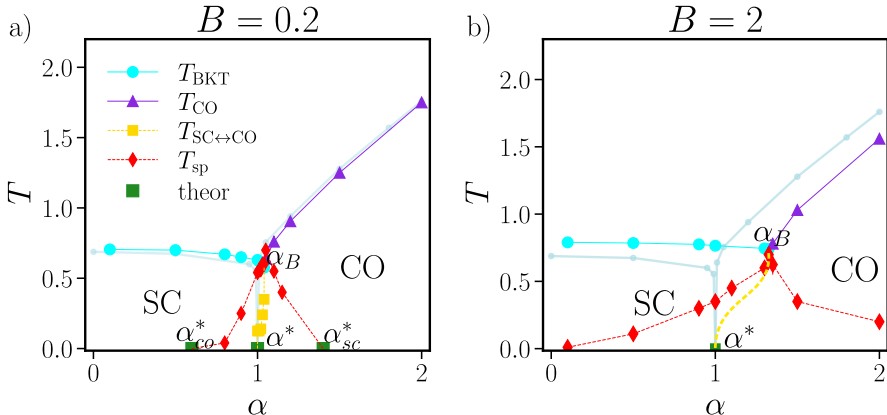

Figure 6: Phase diagram in the $T$ vs. $\alpha$ plane for the competition between SC and CO, modelled with an XXZ model with a barrier height (a) $B = 0.2$ and (b) $B = 2$. The light blue lines refer to the bare XXZ model, for comparison. Cyan dots are the $T_{\mathrm{BKT}}$ points calculated with the BKT scaling law [Eq. (3)]; purple triangles refer to the Ising (CO) transition, for which the $T_{\mathrm{CO}}$ points are found by use of the Binder cumulant $U_N$ [Eq. (5)]; yellow squares locate the first-order transition, while red diamonds indicate the spinodal points; green points at $T = 0$ are calculated analytically; yellow squares and red diamonds in (a) are computed from the effective free energies $F(m_z)$ and the locations of its minima $F_{\mathrm{min}}(T)$, while in (b) they are inferred within the protocol described at the end of Sec. 3.1

$\alpha_B > 1$ and the first-order-line between the charge-ordered and the superconducting state has a positive slope, indicating that entropy slightly favours SC.

The first-order transition line ($T_{SC \leftrightarrow CO}$, marked by the yellow squares) and the spinodal lines ($T_{\mathrm{sp}}$ red diamonds) in the phase diagram are obtained by constructing the effective distribution function $P_{\mathrm{eff}}(m_z)$, as discussed in Sec. 3. We report in Fig. 7a the minima of the free energy $F_{\mathrm{min}}(T)$ as a function of the temperature, for the case $\alpha = 1.04$, where the superconducting state is marked in red and the charge-ordered state in green. The crossing point between the two curves is the first-order critical temperature $T_{SC \leftrightarrow CO}$. We see that, in agreement with our previous discussion, CO is the stable phase at low temperatures, then, with rising the temperature, the system switches to superconductivity and then reaches the disordered state. We predict that in a very clean system close to the $p_{O(3)}$ point this phenomenon of superconductivity stabilized by temperature could be seen.

In panels b-e of Fig. 7, we report the histograms at the temperatures $T = 0.25$, $0.35$, $0.60$, $0.65$, where the distribution of $m_z$ is in turquoise (left axis) and the corresponding free energy $F(m_z)$ is in magenta (right axis). At $T = 0.25$, $F(m_z)$ displays three minima, Fig. 7b, the global ones being at $m_z = \pm 1$ (corresponding to CO). By increasing the temperature, at $T = 0.35$, the three minima become equivalent, Fig. 7c, while at $T \geq 0.35$ the global minimum is at $m_z = 0$ (corresponding to SC). To define the spinodal temperature, at each $\alpha$, we fitted the data $F(m_z)$ in the region around $m_z = 0.5$, and looked for the temperature at which the free-energy curvature changes from downward to upward. It can be seen, by comparing panels (d) and (e) in Fig. 7 how the two minima at $m_z = \pm 1$ disappear when the temperature is increased from $T = 0.60$ (panel d) to $T = 0.65$ (panel e), where the curvature near $m_z = \pm 1$ appears to be flat.

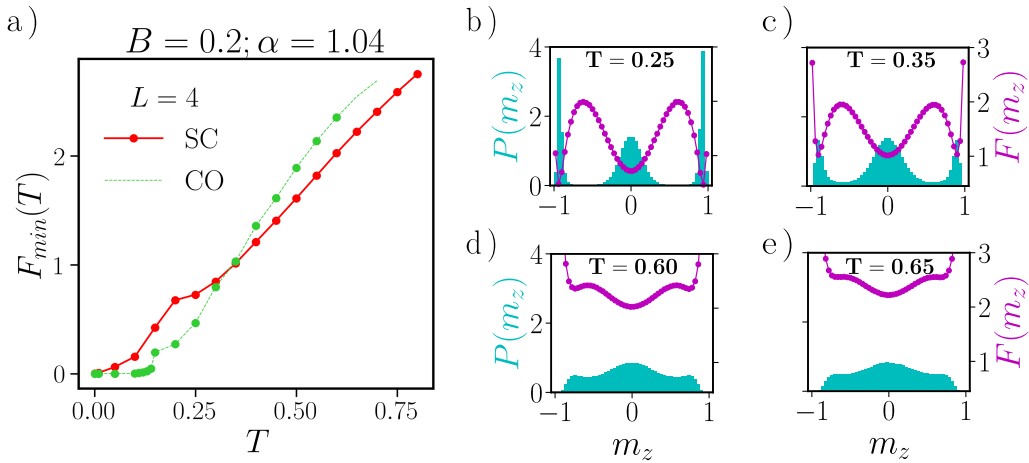

Figure 7: Effective free energies and probability distributions of $m_z$ for a system of linear size $L = 4$ with $\alpha = 1.04$, $b = 0.1$. (a) Local minimum of the free energy as a function of temperature $F_{\min}(T)$. The red curve corresponds to the minimum $F(m_z \approx 0)$ while the green line corresponds to the minimum $F(m_z \approx 1)$. The crossing temperature between the two lines marks the first order transition $T_{SC \leftrightarrow CO}$. (b-e) Effective probability density function $P(m_z)$ (turquoise) and free energy $F(m_z)$ (magenta) for: (b) $T = 0.25$, (c) $T = T_{SC \leftrightarrow CO} = 0.35$. (d) $T = 0.60$, and (e) $T = T_{\mathrm{sp}} = 0.65$. The free energies $F(m_z)$ at each temperature were constructed from the distribution of $m_z$ within $N_{\mathrm{MC}} = 5 \times 10^5$, $\tau_{\mathrm{MC}} = 5 \times 10^5$ and $\tau_{\mathrm{MC}} = 50$.

## 3.4 Phase diagram for $B = 2$

Let us consider now a barrier parameter $B = 2$. The phase diagram in Fig. 6b shows that the BKT line survives for values $\alpha \gg 1$, up to $\alpha_{\mathrm{B}} = 1.325$. The two spinodal points at $T = 0$, calculated according to Eq. (7), are located at $\alpha^*_{\mathrm{CO}} = -3$ which correspond to a reversed interaction in the charge sector and $\alpha^*_{\mathrm{SC}} = 5$ (both outside the displayed range). Again, the BKT and CO points are extracted as discussed in Sec. 3.3: the cyan circles represent the $T_{\mathrm{BKT}}$ temperature, computed with the scaling relation of the superfluid stiffness $J_{\mathrm{s}}$ [Eq. (3)]; and the purple triangles are used to mark the Ising transition temperature $T_{\mathrm{CO}}$, computed from the finite-size scaling analysis of the Binder cumulant $U_N$ [Eq. (5)]. To determine the spinodal points at finite temperature (red diamonds), we rely on the protocol discussed at the end of Sec. 3. The yellow dashed line is a guide to the eye, to sketch the expected first-order transition line connecting the $T = 0$ transition point at $\alpha^*$ (green square) to the bicritical point at $\alpha_{\mathrm{B}}$.

In Fig. 5 we plot the superconducting ($\widetilde{\chi}^{xy}$) and charge-ordered ($\widetilde{\chi}^z$) mean-square magnetization (panels d and e, respectively), and the susceptibility $\chi^z$ (panel f), at different values of the anisotropy parameter, in the range $0.1 < \alpha < 2$, for $B = 2$. As one can see, $\widetilde{\chi}^{xy}$ is significant for values of anisotropy $\alpha \leq 1.325$, well above the isotropic Heisenberg limit. The situation gets reversed as soon as $\alpha \geq 1.35$, where the main response of the system is in the out-of-plane direction (corresponding to CO). As in Sec. 3.3, the susceptibility $\chi^z$ shows precursor peaks of the charge-ordered state found at $\alpha \leq 1.35$, down to $\alpha = 1.3$, where a very broad peak can observe.

The case $\alpha = 1.35$ highlights again the possibility of having a superconducting state stabilized by entropic effects. Indeed, upon cooling, the superconducting mean-square order parameter, light blue curve in Fig. 5c, follows the BKT behaviour lowering $T$, similarly to the curve at $\alpha = 1.325$ (green curve), down to $T = 0.8$. By further lowering $T$

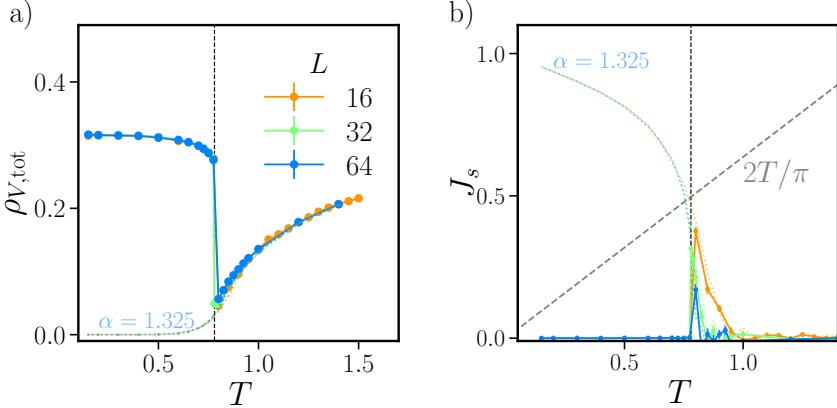

Figure 8: (a) Total density of vortices and antivortices $\rho_{V,\mathrm{tot}}$ [Eq. (4)] as a function of the temperature, for $\alpha = 1.35$, $B = 2$, shows a re-entrant phase as $\rho_{V,\,\mathrm{tot}}$ seems to decay exponentially lowering $T$ down to a temperature $T_{\mathrm{c}}$ where vortices suddenly proliferates. $T_{\mathrm{c}}$, marked with the vertical dashed line, was deduced from the Binder cumulant $U_N$. (b) The same trend is also confirmed by the finite-size effects in $J_{\mathrm{s}}$. Error bars are calculated using the bootstrap resampling method with 100 datasets and blocks of size 100. The case $\alpha = 1.325$, showing typical BKT feature, is plotted in lighter colours as a benchmark.

to $T = 0.775$, $\widetilde{\chi}^{xy}$ drops down by a factor $\sim 500$ and, correspondingly, $\widetilde{\chi}^{z}$ is increased by a factor $\sim 800$. It is worth noting how the peak in $\chi^z$ for this value of anisotropy is still very smeared as for $\alpha < 1.35$, thus leaving no doubt about the nature of the ground state.

To fully describe the properties of the anomalous transition found at $\alpha = 1.35$, we looked at the evolution in temperature of the total density of vortices, given in Eq. (4). The $T$-dependence of $\rho_{V,\,\mathrm{tot}}$, is shown in Fig. 8a. In the BKT scenario, $\rho_{V,\mathrm{tot}}$ is supposed to be exponentially suppressed as the temperature is lowered towards $T_{\mathrm{BKT}}$, as a consequence of the binding of vortex-antivortex pairs. This happens up to $\alpha = 1.325$ (show as a benchmark with light colours in Fig. 8a), where the suppression of $\rho_{V,\,\mathrm{tot}}$ coincide with the appearance of a finite $J_{\mathrm{s}}$ (light colours in Fig. 8b). The $\alpha = 1.35$ curve seems to follow this trend in the high-temperature regime. Crossing the temperature $T \approx 0.8$, a sudden proliferation of free vortices is observed. This indicates an anomalous transition from an almost BKT-like superconducting state at high temperatures, turning into a charge-ordered state below $T \approx 0.8$, in agreement with the trend found in $\widetilde{\chi}^{xy}$ and $\widetilde{\chi}^{z}$. Note that such anomalous behaviour is also detected by finite-size effects in the superfluid stiffness plotted in Fig. 8b. At high temperatures, the paramagnetic phase seems to be on the verge of undergoing a BKT transition, as it is visible from the tails of $J_{\mathrm{s}}$, while instead at $T \simeq 0.76$ (vertical dashed line) the system develops CO and $J_{\mathrm{s}}$ drops to zero.

Finally, let us discuss in more detail the metastable states for $B = 2$. As an example, we report in Fig. 9 the superfluid stiffness, rescaled according to Eq. (3), for $\alpha = 0.5$ (panel a) and the mean-square magnetization $\langle m_z^2 \rangle$ signalling CO, for $\alpha = 2$ (panel b). The dots shown in the plot are computed cooling down the system from a random configuration at a given size $L$ while thick lines stand for the heating up process from the metastable state. As one can see, the jump from the metastable state at low temperature to the ground state is not strongly dependent on the system size $L$.

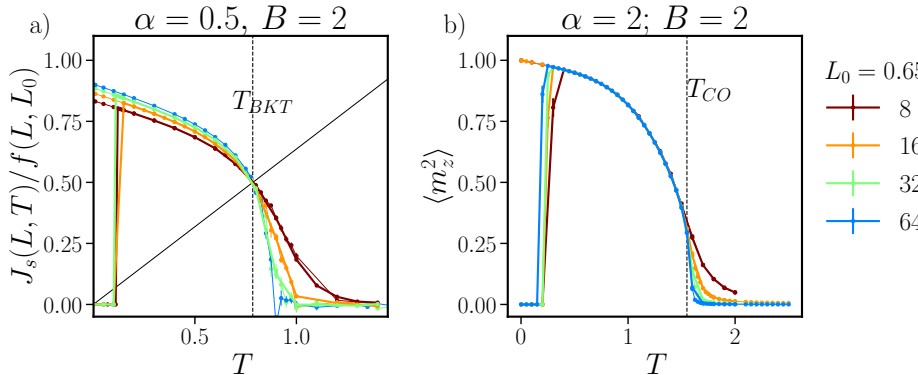

Figure 9: (a) Superfluid stiffness rescaled according to Eq. (3) (in the label of the vertical axis we defined $f(L, L_0) = 1 + [2 \ln(L/L_0)]^{-1}$ for brevity) for $\alpha = 0.5$ and (b) mean-square charge-ordered magnetization $\langle m_z^2 \rangle$ for $\alpha = 1.5$ at various $L$ (color code as indicated in the legend). Thin lines correspond to the usual cooling down protocol; thick lines stand for the results obtained when the system is heated up starting from the metastable state – (a) up, (b) parallel and on the $xy$ plane. Note that the temperature at which the system jumps from the local to the global minimum state is not strongly dependent on the system size $L$. The error bars are calculated using the bootstrap resampling method with 100 dataset and blocks of size 100.

# 4   Dirty system

We now discuss the role of disorder. The localizing effect of impurities is not expected to significantly alter the BKT transition found at $\alpha < 1$ [50]. For $\alpha > 1$, a study at zero temperature has shown that the effect of disorder is to break CO into a polycrystalline state [9, 10, 51]. This can be seen in Fig. 10, where we show low-temperature snapshots ($T = 0.001$) of the MC simulations for increasing $\alpha$. The colour code maps the CO order parameter ranging from $S^z = +1$ (blue), through $S^z = 0$ (no CO, yellow), to $S^z = -1$ (red). We remind the reader that $S^z = \pm 1$ encodes two variants of the CO, e.g., with maxima of the charge density located at two different lattice positions, connected by translational symmetry of the lattice. It was argued in Refs. [9, 10, 51] that at the boundary of such domains CO gets frustrated and FSC emerges. Indeed, as CO fluctuations are enhanced by increasing $\alpha$, the superconducting condensate gradually loses its two-dimensional nature by forming thinner and thinner filamentary structures. As a consequence, as the superconducting cluster get narrower, a smearing of the BKT signatures is expected.

To probe the BKT transition, we monitor the superfluid stiffness $J_s$ and its scaling according to Eq. 3. However, this will not be sufficient in our discussion because of the gradual broadening of the BKT jump of $J_s$, along with the gradual violation of the BKT scaling relation, Eq. (3). Moreover, a substantial fraction of in-plane pseudospins will survive also in the charge-ordered region of the phase diagram, as it is already visible from the snapshots in Fig. 10. Thus, in the dirty system, we will also study the disorder-mediated superconducting correlation function among two pseudospins separated by $\boldsymbol{r}$,

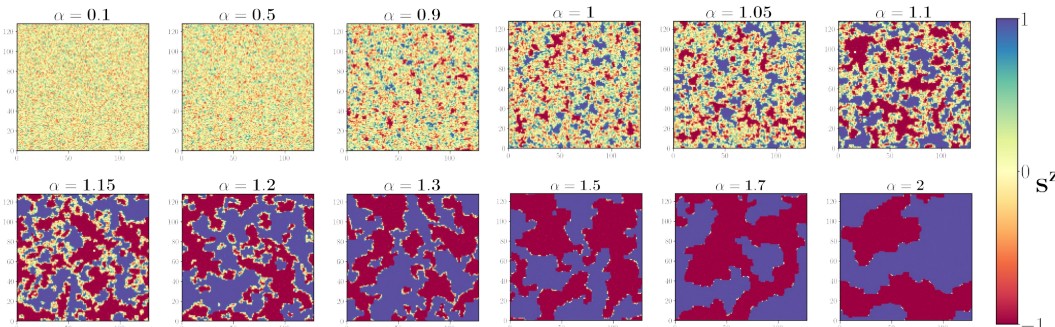

Figure 10: Snapshots of the final MC configuration at $T = 0.001$ for systems of size $L = 128$ and $\alpha = 0.1, 0.5, 0.9, 1, 1.05, 1.1, 1.15, 1.2, 1.3, 1.5, 1.7, 2$. The colour code maps the $S^z$ component of the pseudospin, ranging from $+1$ (blue), to $0$ (in-plane, yellow), to $-1$ (red).

defined as

$$C^{xy}(\boldsymbol{r}) = \overline{\left\langle \sum_{\boldsymbol{R}} \left( S^x_{\boldsymbol{R}} S^x_{\boldsymbol{R+r}} + S^y_{\boldsymbol{R}} S^y_{\boldsymbol{R+r}} \right) \right\rangle} = \overline{\left\langle \sum_{\boldsymbol{R}} \sin \varphi_{\boldsymbol{R}} \sin \varphi_{\boldsymbol{R+r}} \cos(\theta_{\boldsymbol{R}} - \theta_{\boldsymbol{R+r}}) \right\rangle}. \quad (8)$$

Note that the average over many disorder realizations (indicated by the overline) restores spatial isotropy at large distances.

Indeed, as it is well known, one of the hallmarks of the BKT topological phase transition is encoded in the peculiar behaviour of the correlation function:

$$C^{xy}(\boldsymbol{r}) \sim \mathrm{e}^{-|\boldsymbol{r}|/\xi^{xy}}, \qquad T > T_{\mathrm{BKT}}, \qquad\qquad \xi^{xy} = 1/\ln(2T/J),$$

$$C^{xy}(\boldsymbol{r}) \sim \left( \frac{a}{|\boldsymbol{r}|} \right)^{\frac{T}{2\pi J}}, \qquad T \le T_{BKT}, \qquad\qquad \xi^{xy} \to \infty,$$

in the thermodynamic limit, where $J$ is the stiffness at $T = 0$ and $a$ is the characteristic size of a vortex core [52]. The infinite correlation length of superconducting fluctuations $\xi^{xy}$ at $T \le T_{BKT}$ cannot be probed by numerical simulations of a finite system. Instead, in a MC simulation, one has $\xi^{xy} \propto L$. It should also be noted that, as a consequence of the presence of out-of-plane fluctuations, Eq. (8) acquires an extra factor $\sin \varphi_{\boldsymbol{R}} \sin \varphi_{\boldsymbol{R+r}}$ with respect to the standard BKT case, in which $C^{xy}(\boldsymbol{r}) = \overline{\langle \sum_{\boldsymbol{R}} \cos(\theta_{\boldsymbol{R}} - \theta_{\boldsymbol{R+r}}) \rangle}$.

The average in-plane component

$$\overline{\langle \sin \varphi \rangle} = \overline{\left\langle \sum_{\boldsymbol{R}} \sin \varphi_{\boldsymbol{R}} \right\rangle}$$

provides instead a good estimate of the short-range SC still present in the system. It is worth noting that this quantity again does not contain the information about the coherence of the condensate, which is encoded in the $\theta$ variable. We checked, however, that for all values of $\alpha$ we investigated, at least at low temperatures, the spins belonging to the same superconducting cluster are indeed coherent.

Moreover, the charge-ordered state loses now its long-range order so that the resulting magnetization and the corresponding Binder cumulant in Eq. (5), cannot be used to define $T_{\mathrm{CO}}$. For $\alpha > 1$, the system gradually evolves toward a random-field Ising model and the

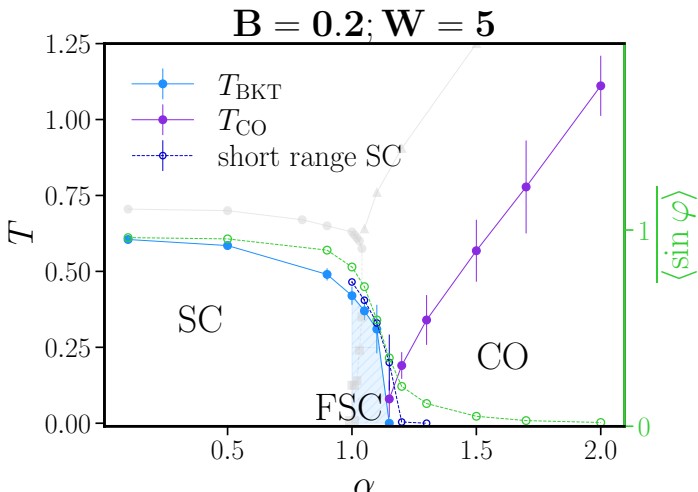

Figure 11: $T$ vs. $\alpha$ phase diagram for $B = 0.2$ and $W = 5$ (light blue, purple and blue symbols and lines, left axis) and average superconducting order parameter $\overline{\langle \sin \varphi \rangle}$ at at $T = 0.001$ (green circles, right axis); the complete temperature dependence of $\overline{\langle \sin \varphi \rangle}$ can be found in Fig. 14b. $T_{\mathrm{BKT}}$ points (cyan) are computed using Eq. 3; $T_{\mathrm{CO}}$ points (purple) are computed from the linear fitting of $1/\xi^z$; short-range SC points (blue) refers to the temperature at which $J_s$ for $L = 16$ crosses the critical line $2T/\pi$. The errorbars are calculated from the standard deviation of independent disorder configurations. The grey symbols show the phase diagram of the clean system ($W = 0$), for comparison.

ground state appears as a rather inhomogeneous landscape, characterized by large charge-ordered puddles. Therefore, we will use the CO correlation function defined as

$$C^{zz}(\boldsymbol{r}) = \overline{\left\langle \sum_{\boldsymbol{R}} S_{\boldsymbol{R}}^z S_{\boldsymbol{R}+\boldsymbol{r}}^z \right\rangle},$$

which is expected to decay exponentially as $\sim \mathrm{e}^{-|\boldsymbol{r}|/\xi^z}$, to characterize the behaviour of the charge-ordered state, using $\xi^z$ as a fitting parameter.

The random-field Ising model in two dimensions has no finite critical temperature [53] and is characterized by a finite low-temperature correlation length that grows exponentially with reducing the strength of the random field [54]. The presence of the barrier term in Eq. (1) further suppresses transverse pseudospin fluctuation at low temperatures and enhances the clustering of up/down charge-ordered regions, even at $\alpha \gtrsim 1$, thereby favouring the polycrystalline behaviour up to a finite temperature $T_{\mathrm{CO}}$.

We estimate $T_{\mathrm{CO}}$ assuming the CO correlation length to behave as $\xi^z(T) \sim (T - T_{\mathrm{CO}})^{-1}$, for $T$ approaching $T_{\mathrm{CO}}$ from above (without getting too close to it), with the index $\nu = 1$ of the clean Ising model [55]. The idea is that, starting from high temperatures, one can follow the critical behaviour of the clean Ising model, down to a temperature at which the system crosses over to the non-critical behaviour of the random-field Ising model and the correlation length saturates to a finite value that determines the typical size of the clusters.

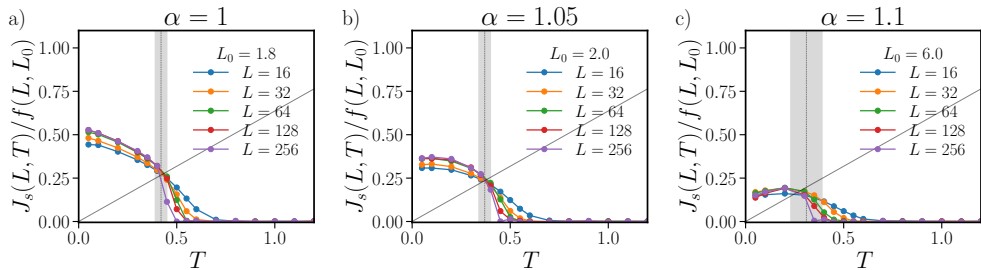

Figure 12: Crossing point of the superfluid stiffness rescaled according to Eq. (3), with the BKT critical line $2T/\pi$ (full black line) at various linear sizes, for $\alpha = 1.1$, $B = 0.2$, and $W = 5$. In the labels of the vertical axis we defined $f(L, L_0) = 1 + [2\ln(L/L_0)]^{-1}$ for brevity.

## 4.1 Phase diagram for $B = 0.2$, $W = 5$

We present our results for barrier height $B = 0.2$ and disorder strength $W = 5$, to explore the effect of disorder in a situation when the first-order transition between the two phases would be nearly vertical in the clean case (see Sec. 3.3).

The phase diagram $T$ vs $\alpha$ is reported in Fig. 11, where the $T_{\text{BKT}}$ points (cyan dots) are calculated using the BKT scaling law of $J_{\text{s}}$, while $T_{\text{CO}}$ (blue) is computed from the fit of $1/\xi^z$ in the temperature range where it exhibits a linear behaviour.

For $\alpha < 1$, the superconducting state is not much affected by disorder, except for a small suppression of the superfluid stiffness (see Appendix A). Indeed, according to the Harris criterion [50], the presence of spatially uncorrelated disorder does not alter the topological phase transition.

Instead, non-trivial features are expected for values $\alpha \gtrsim 1$ where spatially correlated disorder emerges from the interplay between the competition and the presence of impurities. Our two striking results are indeed found on the $\alpha \geq 1$ side of the phase diagram: i) the observation of FSC for $\alpha \gtrsim 1$; and ii) the formation of a polycrystalline charge-ordered phase when $\alpha \gg 1$.

In Fig. 12 we report the rescaled superfluid stiffness $J_{\text{s}}(L, T)$ at various $L$ (the grey line is the critical line $2T/\pi$), following the scaling law in Eq. (3). We find that the BKT scaling law works quite well up to $\alpha = 1.1$, see Fig. 11), despite we observe deviations in the non-universal features of the phase transition. The curves are obtained averaging over $N_{\text{dis}} = 20$ disorder realizations for $L = 16, 32$, $N_{\text{dis}} = 15$ for $L = 64$, $N_{\text{dis}} = 10$ for $L = 128$, and $N_{\text{dis}} = 7$ for $L = 256$. The vertical line and the grey shaded area correspond to $T_{\text{BKT}} \pm \sigma_{T_{\text{BKT}}}$ (the error being calculated as to include the smearing of the jump and the uncertainty on the fitting parameter $L_0$).

It is not surprising that for $\alpha = 1$ (panel a) we still observe a pretty clear jump of the stiffness, although the clustering of small charge-ordered regions in the system can already be observed (see the corresponding snapshot in Fig. 10). In fact, as we showed in Sec. 3.3, in the clean system the potential barrier stabilizes the superconducting state up to $\alpha_B = 1.04$ .

Going towards $\alpha = 1.05$ (panel b) we can still observe a well-defined crossing of $J_{\text{s}}$ with the critical line, whereas in the clean system this value corresponded already to a charge-ordered global minimum of the free energy.

Finally, for $\alpha = 1.1$, we still find a finite superfluid stiffness, and yet the usual BKT scaling relation, Eq. (3), has noticeable deviations as one can realize by scrutiny of Fig. 12c. Rescaling the curves according to Eq. (3) leads to a spread of crossing point with the crit-

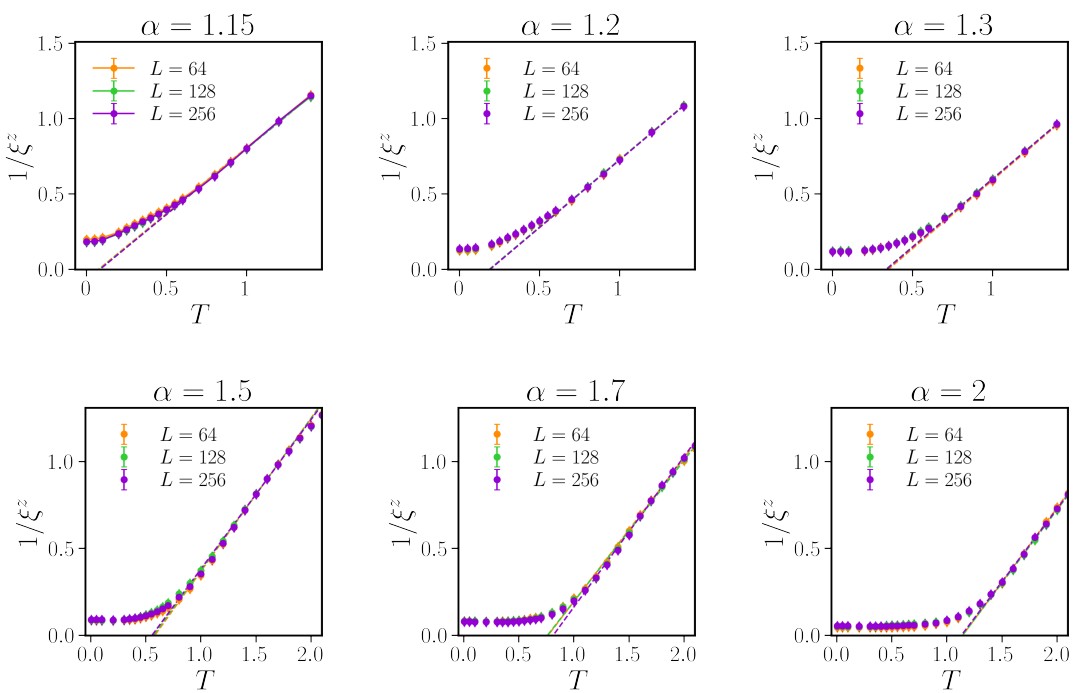

Figure 13: Inverse CO correlation length, $1/\xi^z(T)$, for $B = 0.2$, $W = 5$, different values of the anisotropy $\alpha$, and sizes $L$. The dashed lines are the linear fit of $1/\xi^z$. The errorbars are calculated from the standard deviation $N_{\mathrm{dis}}$ of independent disorder configurations and $N_{\mathrm{dis}}$=15, 10, 10 respectively for $L$ =64, 128, 256.

ical line $2T/\pi$. Notwithstanding that, notice that compared with the unscaled curves shown in Fig. 18c) of Appendix A, the different curves here have a convergence to a small region of temperatures showing approximate scaling. Using the BKT scenario, we obtain $T_{\mathrm{BKT}} = 0.31 \pm 0.08$. This is consistent with the (negative) minimum value of the derivative of the superfluid stiffness with respect to temperature (see Refs. [37,56] and Fig. 18d) in Appendix A). To take into account uncertainties in the definition of the BKT critical temperature, we considered a conservative confidence interval highlighted in gray in Fig. 11c. The difficulties in applying scaling relations in this case can be linked to the emergence of new length scales, presumably related to the geometrical structure of the system, along with the typical sizes of vortex-antivortex pairs in the BKT theory.

It is very interesting that in this regime the stiffness decreases with temperature. This can be seen as a remnant of the entropy-induced superconductivity observed in the clean system, which disfavours superconductivity at low temperatures. Such an anomaly may be measured in samples close to the $p_{O(3)}$ point and is another prediction from this work.

Note that the downward curvature in the low-temperature limit is not the consequence of finite-size effects, since a $256 \times 256$ lattice with periodic boundary conditions provides already a reasonable size to observe reliable thermodynamic quantities.

Let us discuss the CO correlation lengths $\xi^z$, used to define the polycrystalline charge-ordered phase for $\alpha \geq 1.15$ (see purple dots in Fig. 11). The inverse correlation lengths $1/\xi^z$ as a function of temperature are displayed in Fig. 13 for sizes $L = 64$, 128, 256. The dashed lines correspond to the linear fits, and the different colours refer to the system size $L$, as shown in the legend. As one can see, the linear decrease in temperature of $1/\xi^z$ deviates towards a constant plateau when the temperature is lowered below a finite value, the intercept of the linear fit defining the CO critical temperature $T_{\mathrm{CO}}$. Moreover, all

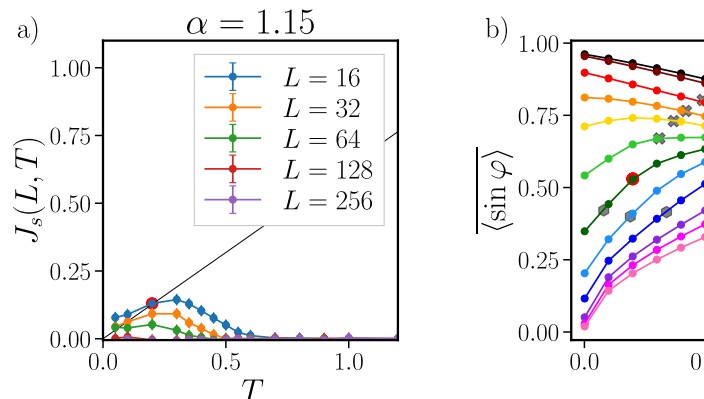
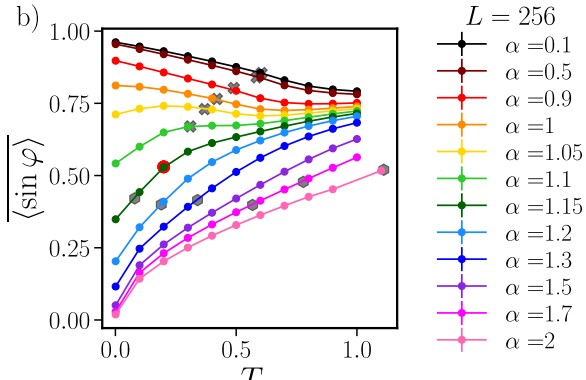

Figure 14: (a) Superfluid stiffness $J_s$ at $\alpha = 1.15$, $B = 0.2$, and $W = 5$, for different system sizes. MC parameters are $N_{\mathrm{MC}} = 2 \times 10^3$, $\tau_{\mathrm{MC}} = 100$, $N_{\mathrm{out}} = 2 \times 10^5$; $N_{\mathrm{dis}}$ are 25 ($L = 16$), 20 ($L = 32, 64$) and 15 ($L = 128, 256$). The errorbars are calculated from the standard deviation of $N_{\mathrm{dis}}$ independent disorder configurations. (b) superconducting order parameter component $\overline{\langle \sin \varphi \rangle}$ as a function of temperature for various anisotropies $\alpha$ at $L = 256$. Crosses are $T_{\mathrm{BKT}}$ points and hexagons are the $T_{\mathrm{CO}}$ points. The red dot highlights the short-range SC in both panels.

the curves show no sign of scaling and the low-temperature saturation value only depends on the anisotropy $\alpha$ (and on the barrier $B$ and on the strength of disorder $W$, if they were allowed to vary), thus signaling the presence of an intrinsic length scale related to the clusters. The downward deviation from linearity at high temperature, observed in the studied temperature interval when $\alpha \geq 1.5$, signals that the system is exiting the critical regime of the clean Ising model with further increasing $T$. We would have observed the same deviation for $\alpha < 1.5$ by looking at higher temperatures.

We conclude this section by observing that the curves in Fig. 13 resemble the behaviour of the full width at half maximum of the CDW peak [proportional to $(\xi^z)^{-2}$] probed in $YBa_2Cu_3O_{7-\delta}$ and $Nd_{1+x}Ba_{2-x}Cu_3O_{7-\delta}$, by means of resonant inelastic X-ray scattering in Ref. [15]. There, the extrapolated $T_{\mathrm{CO}}$ coincides with the temperature at which CO would occur once SC is suppressed by a magnetic field [1, 13], while the saturation at low temperature, in the absence of a magnetic field, signals that CO competes with SC, and SC is more stable. Here, instead, the CO temperature obtained by this criterion near the $O(3)$ point is much lower than the asymptotic value at large $\alpha$. We will come back to this important point in the conclusions.

## 4.2 Short-range and filamentary superconductivity

We discuss now the survival of a filamentary superconducting cluster and the presence of short-range SC in the polycrystalline charge-ordered side of the phase diagram. Whereas in the case $\alpha = 1.1$ it was still possible to define a BKT transition, albeit with a certain degree of uncertainty, when the anisotropy parameter is increased up to $\alpha \geq 1.15$ no $T_{\mathrm{BKT}}$ can be defined from the crossing point. In particular, at $\alpha = 1.15$ one can see from Fig. 14(a) that $J_s$ vanishes already at $L = 128$. The point $\alpha = 1.15$ deserves, however, more attention since it displays both a finite critical temperature $T_{\mathrm{CO}}$ and a short-range coherence of the superconducting cluster, as indicated by the finite value of $J_s$ at $L = 16, 32, 64$. The substantial fraction of superconducting pseudospins can also

be observed by comparing the snapshots in Fig. 10 (yellow component), in particular, those corresponding to $\alpha = 1.1, 1.15, 1.2$. Therefore, for $\alpha \geq 1.15$, despite the fact that the nearly one-dimensional nature of the superconducting cluster does not allow for the binding of vortex-antivortex pairs [57], the finite residual superconducting component can still exhibit some short-ranged stiffness. It is worth noting, once again, that in our coarse-grained model the spacing of the pseudospin lattice $a'$ corresponds to the Josephson scale, i.e., $a' \approx \xi_J \approx 11a$ (see Sec. 1), meaning that $L = 16$ corresponds to about $(16 \cdot 11)^2 \approx 31000$ quantum atoms. Such a large coherent region should have a strong impact in transport properties. While we have not computed the resistivity, it is clear that it will be quite small, as large patches of coherent regions will short-circuit the sample. We speculate that a broad transition should be observed with a large drop of the resistivity to a small but finite value. We thus include in our phase diagram the temperature at which we find short-range SC (blue symbols) in the FSC region of our phase diagram of Fig. 11. Those points indicate the crossing of $J_s$ at $L = 16$ with the universal critical line $2T/\pi$. A finite, even if exponentially small, stiffness is found up to values $\alpha = 1.2$. This behaviour of $T_c \approx 0$ superconductivity is reminiscent of transport experiments in cuprates [9, 23].

In order to get a more quantitative idea of short-range SC, in Fig. 14(b) we show the average in-plane component $\overline{\langle \sin \varphi \rangle}$ as a function of temperature, for different values of $\alpha$. Note that in the standard BKT model, with purely planar pseudospins ($\alpha = 0$), this should be identically equal to one at $T = 0$. However, the presence of out-of-plane (corresponding to CO) fluctuations renormalizes it to a lower value, that decreases with increasing $\alpha$. We indicate with gray crosses $T_{\mathrm{BKT}}$ ($\alpha < 1.15$) and with hexagons $T_{\mathrm{CO}}$ ($\alpha \geq 1.15$) discussed above and presented in the phase diagram (Fig. 11). For $\alpha < 1$, where SC is well described within the BKT scenario and no spatially correlated disorder emerges, $\overline{\langle \sin \varphi \rangle}$ increases quite monotonically with lowering the temperature. As $\alpha = 1$ (in orange) we still observe the monotonic increase of $\overline{\langle \sin \varphi \rangle}$ with decreasing $T$, and from superfluid stiffness computations we know that the system still exhibits quite clear BKT signatures (see Section 4.1 and Appendix A).

For $1 < \alpha < 1.15$, at high $T$ we observe first a slow decrease of $\overline{\langle \sin \varphi \rangle}$ with increasing the temperature followed by an inflection point at $T_{\mathrm{infl}} \gtrsim T_{\mathrm{BKT}}$. This range of the control parameter $\alpha$ lies inside the region of the phase diagram that we labelled with FSC in Fig. 11. We stress again that up to $\alpha < 1.1$, it is still possible to define the BKT temperature from the jump of the superfluid stiffness, which is smeared out but still clearly visible. For $\alpha = 1.1$ (light green), instead, the BKT scaling law starts showing deviations, and we observe a downturn of $\overline{\langle \sin \varphi \rangle}$ at $T_{\mathrm{down}} < T_{\mathrm{BKT}}$. This may be related again to the entropically favoured SC of the clean case, which might also be the cause of the downturn of $J_s$ at very low temperatures.

The curve for $\alpha = 1.15$ (dark green) highlights again an interesting crossover scenario, which presents a filamentary pattern, clearly visible in the snapshots, but no long-range stiffness [see Fig. 14a]. In this case, the decrease at high temperature follows a behaviour similar to the one found for $\alpha = 1.1$, but with no inflection point, down to $T = 0.2$ (indicated with a red dot). By further lowering the temperature, $\overline{\sin \varphi}$ becomes steeper. The absence of an inflection point in $\overline{\sin \varphi}$ might be a proxy that the entropically favoured superconducting state is now suppressed by the large CO fluctuations, although a small $J_s$ survives at finite $L$, becoming exponentially small with increasing the size. In fact, by comparing $\overline{\sin \varphi}$ for $\alpha = 1.15$ with the corresponding $J_s$, one can observe that short-range SC is still present ($L < 128$). Note that the curve for $L = 16$ of Fig. 12 has a maximum for $T = 0.3$, then decreases by further lowering $T$, crossing the critical line $2T/\pi$ at T=0.2 (red dot). We point out that at the lowest temperature $T = 0.001$ a substantial superconducting residue survives $\overline{\langle \sin \varphi \rangle} = 0.35$, presenting phase coherence.

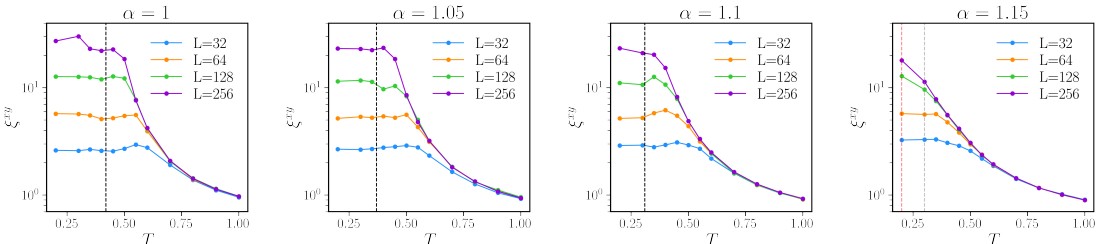

Figure 15: superconducting correlation length $\xi^{xy}$ as a function of temperature for $L = 32, 64, 128, 256$ at various anisotropy parameter $\alpha = 1, 1.05\,1.1, 1.15$. The vertical dashed black lines signal the critical temperatures $T_{\mathrm{BKT}}$, the dashed red line in the panel $\alpha = 1.15$ at $T = 0.2$ is the same temperature indicated (also in red) in Fig. 14; the grey dashed line at $T = 0.3$ correspond to the maximum of $J_{\mathrm{s}}$ for $L = 16$.

Even increasing the anisotropy up to $\alpha = 1.2$, the superconducting fraction is still about 20%. We thus include $\overline{\langle \sin \varphi \rangle}$ at $T = 0.001$ and $L = 256$ in our phase diagram in Fig. 11 (right axis, in green) to stress out the presence of a macroscopic superconducting residue, that can show signatures in transport experiments even if it lacks long-range coherence.

In order to investigate the role of superconducting phase fluctuations in this crossover filamentary state, we analyze the correlation length $\xi^{xy}$. In Fig. 15 we present $\xi^{xy}$ for $\alpha = 1, 1.05, 1.1, 1.15$ and $L = 32, 64, 128, 256$, as found by fitting the correlation function in Eq. (8). For $\alpha < 1.15$, we find that $\xi^{xy} \sim L$, thus following the expected BKT scenario, thereby justifying the BKT analysis discussed above. Note that the black dashed lines marks $T_{\mathrm{BKT}}$ found from the crossing of $J_{\mathrm{s}}$ (Fig. 12). For $\alpha = 1.15$, although $J_{\mathrm{s}}$ vanishes at $L = 128$, we can still observe BKT-like features of $\xi^{xy}$: the saturation value at low temperatures is in fact increasing with $L$, with some slowing down for $L = 256$. Again, we mark the temperature $T = 0.2$ with a dashed red line: this temperature corresponds to the maximum of $J_{\mathrm{s}}$ found at $L = 16$ and to the change of slope in the decrease of $\overline{\langle \sin \varphi \rangle}$. The behaviour of $\xi^{xy}(T)$ at this temperature seems to suggest the occurrence of an "avoided" superconducting state, reminiscent of the findings in transport experiments [9, 23].

# 5 Three-dimensional phase diagram

Starting from our detailed analysis of the two-dimensional model, we can make a comparison between the theoretical phase diagram Fig. 11 and the experimental phase diagrams of the CO-SC competition of Fig. 2. The most striking difference is that the CO temperature is strongly suppressed near the $p_{O(3)}$ point, both when compared with experiments and with the clean case. Also, FSC does not develop an evident foot in the CO region, although local SC regions are present. To a large extent, both deficiencies can be ascribed to the low dimensionality of the model. To show this, we compute a three-dimensional phase diagram assuming an interlayer coupling both for the CO and the SC phases.

As it is known from studies of the layered Heisenberg model, the transition temperature can be estimated from the superconducting and CO correlation length [58] solving the following equations,

$$[\xi^{xy}(T_{SC}^{3D})]^2 J_{\perp}^{xy} = T_{SC}^{3D}, \qquad [\xi^{z}(T_{CO}^{3D})]^2 J_{\perp}^{z} = T_{CO}^{3D} \qquad (9)$$

meaning that the interlayer energy associated with a correlated region of area $(\xi^{xy,z})^2$ is of the order of the critical temperature.

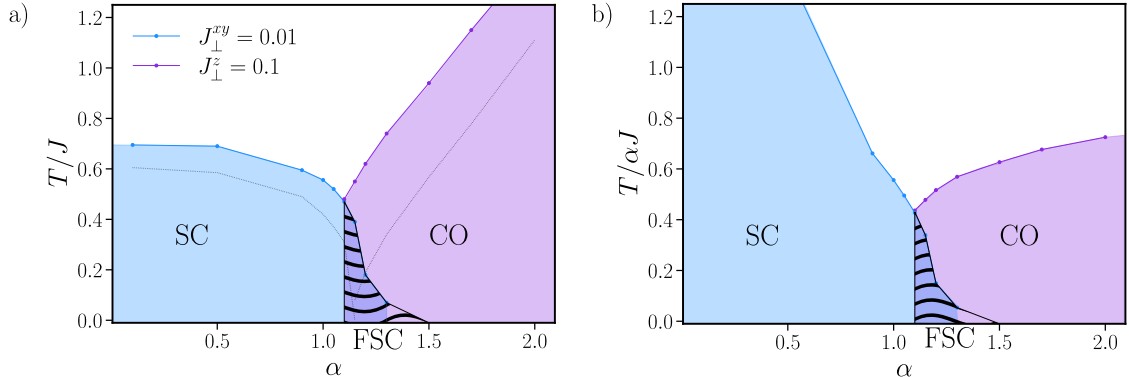

Figure 16: (a) CO-driven and (b) SC-driven phase diagrams. Critical temperatures are normalized with respect to $J$ and $\alpha J$, respectively. The 3D critical temperatures are calculated from the correlation length according to Eq. (9) with $J_\perp^{xy} = 0.01$ and $J_\perp^z = 0.1$. We specify that the FSC point in $\alpha = 1.5$ is extrapolated from the vanishing of the in-plane superconducting component (see purple curve in Fig. 14b). For comparison, the grey lines in panel a) refer to the two-dimensional system.

The superconducting and charge-ordered interlayer couplings, $J_\perp^{xy}$ and $J_\perp^z$ respectively, are not known. In view of the quasi 2D nature of cuprates we take $J_\perp^z = 0.1$ and $J_\perp^{xy} = 0.01$. The much smaller value of the superconducting coupling with respect to the charge-ordered one is justified by the fact that CO is coupled by the long-range Coulomb interaction while SC is coupled by Josephson tunnelling through the insulating layers, and one expects a large difference between these two scales. For the rest, these parameters are rather arbitrary, but the qualitative form of the phase diagram is not expected to be sensitive to the precise value of the couplings.

Fig. 16(a) shows the resulting phase diagram. The two-dimensional lines are showed with dotted grey lines for comparison. Panel (a) follows our previous convention of measuring energy and temperature in units of the superconducting scale $J$ so that the scale of the CO state changes with $\alpha$. This corresponds to the CO-driven transition mentioned in the introduction. In panel b, by simply rescaling our energy units, we derive the phase diagram for the SC-driven transition. Here, the CO energy scale is, by definition, constant.

We see that indeed the three-dimensional phase diagram bears a strong resemblance with the experimental phase diagrams for the SC-driven case (Fig. 2a,b) and CO-driven case (Fig. 2c). Now the bicritical point is at a temperature of the order of the ordering temperature and the FSC foot extends more in the CO region.

# 6    Conclusions

We used Monte-Carlo simulations to solve a statistical mechanics model of a two-dimensional system presenting competition between SC and CO, both in the absence and in the presence of quenched disorder. We computed thermodynamic quantities, correlation functions, and thermodynamic phase diagrams.

In a clean system, the competition mechanism generates metastability regions in the phase diagram, bounded by two spinodal lines and encompassing the first-order phase transition line. As the temperature increases, the region of metastability shrinks to a single point, which coincides (within numerical accuracy) with the bicritical point, where

the charge-ordered, superconducting and disordered phases meet. The first-order line separating the charge-ordered phase and the superconducting phase is rather steep for low values of the barrier height ($B = 0.2$), indicating that the two phases have similar entropy, as one can check using the Clausius-Clapeyron relation. We can thus make a comparison with the case of $^4$He [30, 31], where the almost vertical line separating the solid and superfluid phases led to the hypothesis that superfluidity was a low-entropy phase, as a crystal, fuelling explanations based on condensation in momentum space rather than in real space.

A closer inspection shows that the first-order line is not exactly vertical, and a re-entrance appears, thus showing that near $\alpha \gtrsim 1$ one can make a transition from the charge-ordered phase to the superconducting phase by increasing the temperature. This means that the superconducting phase has actually slightly higher entropy than the charge-ordered phase. A posteriori, this result is reasonable as the charge-ordered state has two gapped transverse modes while the superconducting state has one gapped mode and one Goldstone mode. Thus, just considering low-laying excitations near $T = 0$, it is reasonable that the superconductor can have larger thermal fluctuations and entropy. Interestingly, the re-entrant behaviour of the superconducting phase is also reminiscent of the phase diagram of $^4$He, in which a range of pressures is found where the solid $^4$He, if heated, transits to its superfluid state before becoming a simple liquid. In $^4$He, however, this happens in the high-temperature part of the phase diagram, while here we observe it at low temperatures. In fact, in the low-temperature region, the slope of our phase diagram and the one of $^4$He have opposite sign. We speculate that this qualitative difference is due to the fact that in our case the charge-ordered state has no Goldstone modes while in the case of $^4$He the crystal has sound (Goldstone) modes.

The metastability regions and the first-order transition line disappear when quenched disorder is considered, giving rise instead to a phase-separated region where FSC appears. Indeed, moving from the SC to the CO regime we find the gradual disappearance of the two-dimensional superconducting phase towards a polycrystalline charge-ordered phase with the tuning of the anisotropy parameter $\alpha$. As the BKT signatures disappear, one-dimensional-like superconducting patterns still survive inside the charge-ordered phase.

In Refs. [8–10] it was already proposed that disorder may have a peculiar effect in the coexistence region discussed above, turning the metastable superconducting state into a stable state, where FSC is topologically protected at the boundaries between different charge-ordered domains, in agreement with the tentative phase diagram proposed for cuprates in Ref. [8]. Such a phase diagram was purely based on the peculiarities of the resistance curves as a function of the temperature, with varying magnetic field and doping, and showed that SC can develop at low temperatures even when at high temperature the system is well inside the charge-ordered region of the phase diagram. In this work, we provided a solid background to the above scenario, showing that within our minimal model for the competition between CO and SC, Eq. (1), a random magnetic field has exactly the effect of promoting the fragmentation of the charge-ordered state into domains exhibiting the two different realizations of CO. In the domain wall they frustrate each other resulting in the stabilization of the superconducting state. Once this FSC is suppressed (by increasing the temperature and/or the non-thermal parameter $\alpha$), only polycrystalline CO remains. The polycrystalline CO is characterized by large puddles with different realizations of CO, and it resembles the complex landscape of charge-ordered domains experimentally observed in cuprates [59].

The filamentary superconductivity foot we find in the two-dimensional phase diagram (Fig. 11) is relatively small compared with experiments (Fig. 2). Also, the CO temperature is strongly suppressed close to the $O(3)$ point in the presence of quenched disorder. Both

features are cured considering an interlayer coupling, yielding a phase diagram nicely resembling experiments, both in the CO- and SC-driven case.

Very near the $O(3)$ point in the dirty case, we find that the superfluid stiffness has a non-monotonous behaviour as a function of $T$ with a maximum at intermediate temperatures (Fig. 12c). It would be very interesting to observe experimentally this effect as it would be a signature of entropically favoured superconductivity.

In our model, up and down pseudospins encode only two possible realizations of CO corresponding to a checkerboard pattern in a bipartite lattice. This is a simplification of cuprates where, for example, non-magnetic charge stripes with periodicity four have four CO variants for each orientation, yielding 16 possible "colours" of CO patterns. Still, our simplified "two-colour" model captures many subtleties of the phase diagram.

The presence of some one-dimensional-like superconducting patterns persisting on the CO side of the phase diagram can indeed have striking effect on the macroscopic observable, such as specific heat [7] or spin susceptibility [60], and particularly on transport measurements [8, 23, 24].

# Acknowledgements

We acknowledge stimulating discussions with M. Grilli, B. Leridon, F. Ricci-Tersenghi.

**Funding information**    We acknowledge financial support from the University of Rome Sapienza, under the projects Ateneo 2020 (RM120172A8CC7CC7), Ateneo 2021 (RM12117 A4A7FD11B), Ateneo 2022 (RM12218162CF9D05), from the Italian Ministero dell'Università e della Ricerca, under the Project PRIN 2017Z8TS5B, PRIN 20207ZXT4Z, from PNRR MUR project PE0000023-NQSTI, and from CNR-CONICET project "New materials for superconducting electronics".

# A    Superfluid stiffness

For the sake of completeness, we show the superfluid stiffness and their BKT critical jump for all the values of $\alpha$ considered to construct our phase diagram (Fig. 11). In Fig. 17 it is possible to observe the validity of Harris criterion when addressing the BKT transition for $\alpha < 1$, where the disorder leaves $J_s$ almost unaffected. The only appreciable effect relies in the suppression of both the saturating value of $J_s$ for $T \to 0$ (see panels a, b and c), which is lowered to 0.75 for $\alpha = 0.9$ while the critical temperature is only very slightly decreased. This can be appreciated looking at panels d, e, and f, where we show the relative crossing points with the universal critical line $2T/\pi$, indicating with a vertical line the corresponding $T_{\text{BKT}}$. A first consequence of the random field is indeed visible in the smearing of this crossing at $\alpha = 0.9$, highlighted in grey.

In panels a, b and c of Fig. 18 instead we present the superfluid stiffness in the filamentary region of our phase diagram, namely $\alpha = 1, 1.05, 1.1$. The suppression of $J_s$ caused by the presence of the correlated disorder emerging is much more visible here. In particular, we highlight the fact that, while for $\alpha = 1, 1.05$ the scaling law still produces efficient results (see panels a and b of Figs. 17 and 12), this does not seem to be the case for $\alpha = 1.1$ ((see panels a and b of Figs. 18 and 12). However, the extrapolated $T_{\text{BKT}}$ is consistent with the minimum found for its derivative $\partial J_s/\partial T$ [37, 56]. This is shown in panel d where the vertical line is at $T_{\text{BKT}}$ and the grey area highlights the error considered.

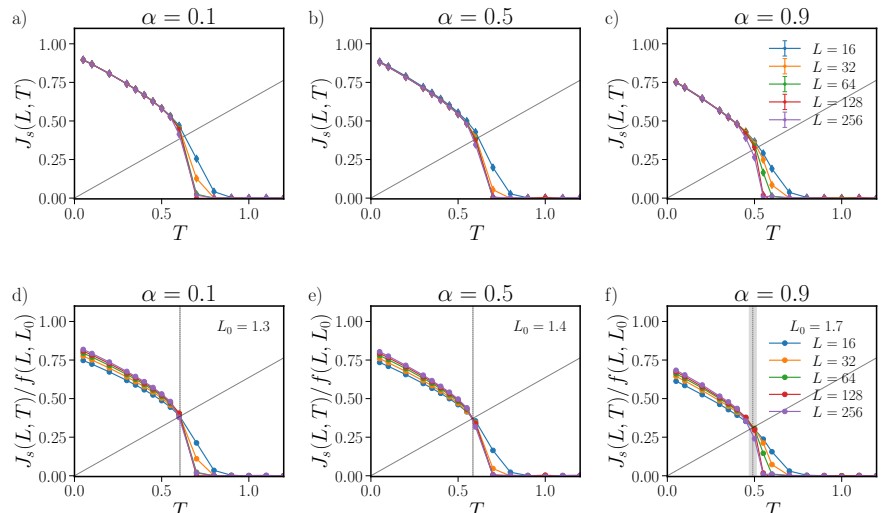

Figure 17: (a) Superfluid stiffness and (b) rescaled superfluid stiffness ($f(L, L_0) = 1 + [2 \ln(L/L_0)]^{-1}$, see Eq. 3) for $\alpha = 0.1, 0.5, 0.9$. Errorbars in panel b refers to the standard deviation computed on different indipendent disorder realizations. The vertical grey lines indicates $T_{\mathrm{BKT}}$, black lines are the universal critical line $2T/\pi$.

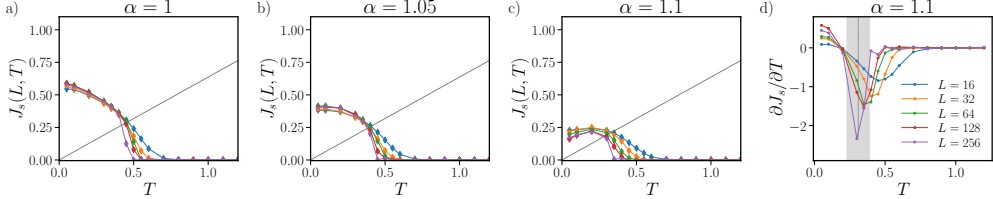

Figure 18: Unscaled superfluid stiffness with errorbars for (a) $\alpha = 1$, (b) $\alpha = 1.05$, (c) $\alpha = 1.1$ and (d) its first derivative with respect to the temperature $\partial J_s/\partial T$. The errorbars in panels a, b and c are calculated from the standard deviation of independent disorder configurations. In panel d the vertical line and the grey shaded area indicates $T_{\mathrm{BKT}}$ with its error, extracted using the BKT scaling law and showed in Fig. 12.

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
