# Peer review of "Thermodynamic phase diagram of the competition between superconductivity and charge order in cuprates"

_SciPost Physics_

## Round 2 · Referee Report · Anonymous (Referee 1) · 2023-9-7

Strengths

this is a carefult classical Monte-Carlo study of the classical anisotropic Heisenberg model (XXZ model) with an effective barrier potential term and a random field mimicking disorder.

Weaknesses

The authors spend quite some time to convience the reader that the model they consider should capture the phase diagram of superconducting cuprates and even put this in the title. There is a series of assumptions to make such a claim and the model described above cannot explain the physics of layere cuprates. In my opinion this model is an oversimplification of the physics of the cuprates.

Report

  • I think the authors should clearly put in the title the model they consider, namely XXZ model. I think the results of this paper are beyond the interest of the high-Tc community and one should catch those by appropriately changing title. Discussing the connection to the cuprates they have to make very clear the limitation of the model.
  • in my humble opinion the model the authors consider has nothing to do with the physics of cuprates. All these mapping from the negative to attractive Hubbard model and then using pseudospins can be done but after the mapping performed there is no space for the true magnetic fluctuation dominated physics. Whereas in cuprates CO and superconductivity (which is by the way d-wave while in their case it is s-wave) compete under the umbrella of the strong magnetic fluctuations. So this means there is a strong player (bosonic 'true spin fields') which the authors completely ignore anf they will change the phase daigram and the details of the competition between CO and superconductivity.

Requested changes

-Siginificance: title need to be changed appropriately, the authors study XXZ model with disorder

  • Vailidity: the descrepancy between the model they consider (CO and s-wave supercondictivty) should be contrasted with the actual cuprate physics (CO and d-wave in the presence of strong magnetic fluctuations)

  • validity: ok
  • significance: ok
  • originality: ok
  • clarity: ok
  • formatting: good
  • grammar: good

Author:  Giulia Venditti  on 2023-10-31  [id 4085]

(in reply to Report 1 on 2023-09-07)
Category:
answer to question
reply to objection

We thank the reviewer for their careful reading and positive review of our work.
Please, find enclosed the PDF file with our point-by-point answer to all the issues raised.

On behalf of all authors,
Dr Giulia Venditti

Attachment:

response_report1.pdf

---

## Round 2 · Referee Report · Anonymous (Referee 2) · 2023-10-4

Strengths

1- Rigorous investigation of generalized XXZ model by Montecarlo 2- Thorough explanation of all the criteria and approaches used in the numerical analysis 3- Interesting final results and general discussion of the phase diagrams

Weaknesses

1- The phenomenological nature of the approach, in connection with the Cuprates, should be acknowledged or discussed further 2- slightly lengthy in some parts 3- clarity of figures might be improved

Report

The manuscript provides a detailed investigation, by means of classical Montecarlo, of a generalized XXZ (anisotropic Heisenberg) model. Besides in-plane vs out-of-plane anisotropy, this includes a potential barrier separating the competing phases which are studied (order in or out of plane) and a random magnetic field.
The study is very careful and appropriate caution is used (and very well explained) to handle finite-size effects, meta-stability and spinodals, and equilibration. Phase diagram are traced and discussed, and in particular the influence of the random field is addressed.

The results are used to discuss the competition of charge-ordering and superconductivity in the cuprates. The parallel between the final phase diagram is apparent, thus I deem this description relevant. However a focus should be put on the phenomenological nature of this analysis.
Indeed, the studied model describes emergent classical degrees of freedom, which are linked to the microscopic physics of cuprates through a series of nontrivial assumptions: the attractive Hubbard model is already a qualitative approach to cuprates, and albeit the mapping in a repulsive model is rigorous, the large U limit of the latter is not obviously justified (and this is not done in the text), while the justification for its classical solution and its implications are just touched upon.
Furthermore, there are three parameters in the model which are explored and/or adjusted (reasonably indeed, but their values are not linked to the cuprates in the text).
Finally the phase diagram wanted features of having a sizeable T_CO at alpha~1 and a “foot” of superconductivity at low temperatures for alpha>~1 are crucially obtained only when including the inter-layer couplings and for a chosen set of values.
Again, all these assumptions look reasonable, but the large number of parameters and the specificity of the approach make this a successful phenomenological description (providing indeed an insight of some aspect of the physics) rather than a compelling simplification of the microscopic physics.

I recommend publication but suggest some tuning of the narrative.

Requested changes

The aforementioned aspect of ad hoc and phenomenological approach should be better highlighted, from the outset (for example in the abstract and introduction).

Not a requested change, rather a suggestion. The article is quite long. Some of the thorough explanations are very welcome indeed, but some parts feel quite slow insted. For instance the B=0.2 diagram of the clean system is practically identical to the known one with B=0. I understand that in this part some concepts used in the subsequent sections are pedagogically introduced, but I wonder if some of this can be moved to the appendix in order to shorten the main narrative.

The figures are at times too small (indeed pdf figures can be zoomed in without quality loss, but if they can be made more readable at the page scale it is better, and in a printed version the information can be hardly readable). Examples: -in both Fig. 6a and in Fig.11 the points around alpha=1 are hard to visually disentangle. Fig. 11 would benefit simply from being text-wide. - The snapshot in Fig 10 and the data in the small panels in Figs. 12, 13 and 15 are too small. Even if their main message is well conveyed at this scale, in order to actually see the data one has to zoom in.
Where possible their readability should be improved.

Typos: - pag.12: "bicrtical" should be "bicritical" - pag.15: Ref. to Sec. 3 should be to Sec. 3.3 - pag 15: Ref. to Fig5c should be to Fig.5f - pag 20: Ref to "blue" in Fig.11 should be to “purple” Section 5. Reference to a "three-dimensional phase diagram" (both in the text and the title) should rather be to a "phase diagram of a three-dimensional system" or something similar.

  • validity: high
  • significance: high
  • originality: high
  • clarity: top
  • formatting: good
  • grammar: excellent

Author:  Giulia Venditti  on 2023-10-31  [id 4086]

(in reply to Report 2 on 2023-10-04)
Category:
answer to question
reply to objection

We thank the reviewer for their careful reading and appreciation of our work.
We provide a point-by-point answer to all the issues raised by the referee. Please, see the attached pdf file.

On behalf of all authors,
Dr Giulia Venditti

Attachment:

response_report2.pdf

---

## Editorial Decision

resubmitted